# Latent Paraphrasing: Perturbation on Layers Improves Knowledge Injection in Language Models

**Minki Kang**[1,2]  **Sung Ju Hwang**[2]  **Gibbeum Lee**[1]  **Jaewoong Cho**[1]

[1]KRAFTON, [2]KAIST

zzxc1133@krafton.com, sjhwang@kaist.ac.kr, {pirensisco, jwcho}@krafton.com

## Abstract

As Large Language Models (LLMs) are increasingly deployed in specialized domains with continuously evolving knowledge, the need for timely and precise knowledge injection has become essential. Fine-tuning with paraphrased data is a common approach to enhance knowledge injection, yet it faces two significant challenges: high computational costs due to repetitive external model usage and limited sample diversity. To this end, we introduce LaPael, a latent-level paraphrasing method that applies input-dependent noise to early LLM layers. This approach enables diverse and semantically consistent augmentations directly within the model. Furthermore, it eliminates the recurring costs of paraphrase generation for each knowledge update. Our extensive experiments on question-answering benchmarks demonstrate that LaPael improves knowledge injection over standard fine-tuning and existing noise-based approaches. Additionally, combining LaPael with data-level paraphrasing further enhances performance.

## 1 Introduction

Pre-trained Large Language Models (LLMs) encode extensive factual information from their training data, enabling them to answer factoid questions such as "Who is the director of Dune: Part Two?" [4, 32]. However, knowledge in LLMs is static, which can lead to outdated information as real-world knowledge evolves. Additionally, LLMs often lack specificity for specialized or private domains. To address this, it is common practice to fine-tune LLMs with updated or domain-specific documents, keeping the model's knowledge up-to-date and enhancing expertise in particular domains [14, 17, 19].

However, does fine-tuning LLMs on a single document allow them to fully internalize its knowledge? Even in pre-training, Kandpal et al. [20] found that LLMs cannot perfectly learn all the information in the training data, particularly long-tail knowledge that appears rarely or only once. Existing work [33] has shown that this issue persists with fine-tuning and suggested that data augmentation, such as paraphrasing, is a simple yet effective way to enhance knowledge injection. As shown in Figure 1, fine-tuning with paraphrases enhances knowledge injection, as evidenced by improved Question-Answering (QA) task performance.

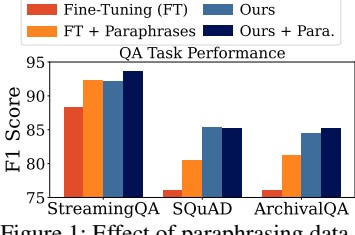

Figure 1: Effect of paraphrasing data in knowledge injection.

While data-augmented approach via paraphrasing is effective for knowledge learning, it has two main limitations: (1) **High computational cost:** Generating high-quality paraphrases requires significant computational resources. As shown in Figure 2, paraphrasing models such as LLMs [5, 7, 11, 58] need to repeatedly generate paraphrases for each document with the new incoming knowledge. This leads to higher costs as the number of documents being learned continually increases; and (2) **Limited**

38th Conference on Neural Information Processing Systems (NeurIPS 2024).

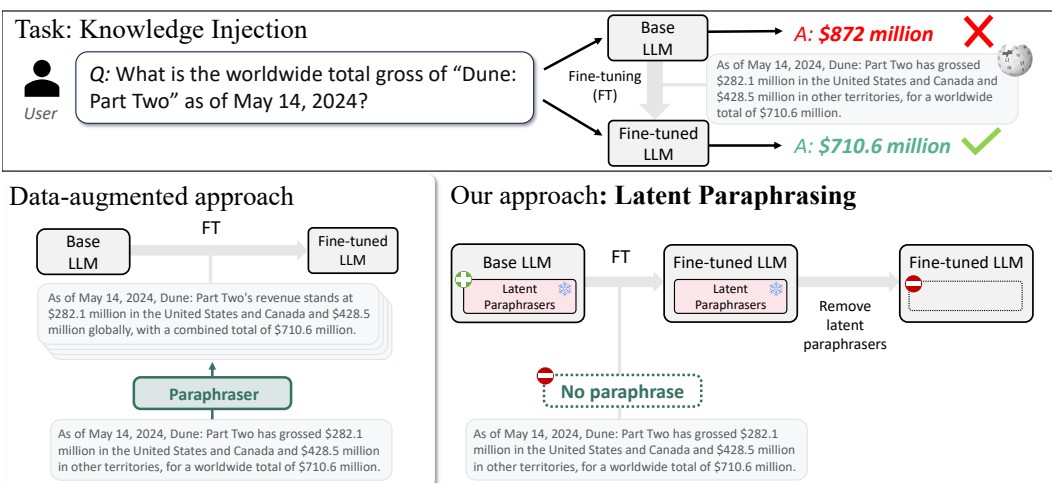

Figure 2: **A conceptual illustration of the proposed approach**. On the left, we show the existing method of knowledge injection by paraphrasing each document for data-level augmentation. On the right, we present the conceptual illustration of LaPael with *trained* latent paraphrasers. Unlike the method on the left, LaPael can eliminate the need for users to repeatedly paraphrase using LLMs once latent paraphrasers are trained.

**diversity in augmented data:** Although LLMs can produce varying high-quality paraphrases by sampling from the generative distribution, the diversity of the generated text is limited, resulting in a narrow range of augmented samples at the discrete data level. One way to overcome these issues is to introduce noise into the token embedding. However, existing works [16, 57] do not consider the text semantics when they perturb the latent features of LLMs with randomly generated noise.

To address these issues, we take a distinct approach using an input-dependent noise generator named "latent paraphraser" learned from the paraphrases. Specifically, this function perturbs early layers to augment LLMs at the latent level while preserving the meaning of the text. To optimize the latent paraphraser, we start by generating paraphrases of the documents. Then, we train the latent paraphrasers to ensure that the latent distribution of the LLMs with the original sentence is close to the latent distribution with the paraphrased sentences. Once training is done, we can transfer the latent paraphrasers to the documents from any domain that contains new knowledge. We refer to our method as **La**tent **Pa**raphrasing of Language Mod**el**s (**LaPael**), as it learns the paraphrasing of text data at the latent level.

We validate our approach on diverse question-answering benchmark datasets [38, 27, 51] designed to evaluate knowledge injection. These benchmarks involve fine-tuning LLMs on documents that contain the knowledge required to answer the questions in the datasets. Our results show that LaPael significantly improves knowledge injection performance compared to standard fine-tuning. Moreover, LaPael outperforms fine-tuning with paraphrases, demonstrating that LaPael alone is sufficient for data augmentation in knowledge injection scenarios, as illustrated in Figure 2. As shown in Figure 1, we further find that using LaPael in combination with paraphrases further enhances performance, providing complementary benefits to data-level augmentations. Finally, LaPael surpasses existing noise baselines [16, 57], highlighting the importance of learning noise for effective augmentations.

Our contributions are as follows:

- We introduce **LaPael**, a new method that applies learned perturbations to the layers of LLMs to enhance knowledge injection, addressing the limitations of data augmentations and noise baselines.
- We validate LaPael using diverse question-answering benchmark datasets, demonstrating a significant improvement in knowledge injection performance compared to standard fine-tuning.
- Our results show that LaPael not only outperforms fine-tuning with paraphrases but also complements it, providing additional benefits when used together, surpassing the performance of existing latent noise-based methods.

## 2 Related Work

**Knowledge of Large Language Models**   Large Language Models (LLMs) store vast amounts of factual knowledge in their pre-trained parameters [36, 44]. The straightforward way to extract the

knowledge of LLMs is to ask the question that requires factual knowledge [43, 58]. Through asking questions, Kandpal et al. [20] have found that LLMs cannot perfectly memorize the entire knowledge in the pre-training corpora, especially for knowledge that appears rarely or only once. To make LLMs answer the question requires under-represented or new knowledge, previous works have clustered into two different solutions. The first one is retrieval-augmented methods [26, 39, 42] that retrieve knowledge from an external knowledge base and input the retrieved knowledge alongside the question into LLMs. The second one is fine-tuning [12, 17] where the parameters of pre-trained LLMs are continually updated by fine-tuned on the document containing knowledge in an unsupervised way as in pre-training [37]. In our work, we focus on improving the fine-tuning-based solution, as storing new knowledge in the parameters of LLMs is efficient since we can reduce the length of the input prompt and do not need any extra module or memory in the deployment time [6].

**Knowledge Injection in LLMs** In this work, knowledge injection in LLMs denotes fine-tuning LLMs on the set of documents to inject new or under-represented knowledge into LLMs [33, 17], different from another task of injecting *symbolic knowledge* (e.g., knowledge graph) into LLMs [55, 54]. Among previous works, CaMeLS [14] has introduced a meta-learning method for learnable loss scaling function that improves knowledge injection. As a concurrent work, MAC [45] has proposed using the memory of amortized context is highly effective in a knowledge injection. However, both methods have drawbacks like high computational costs for bi-level optimization or the need for additional modules and memory. Recent works [33, 58] have shown that data augmentation which paraphrases the knowledge-containing sentences helps language models memorize knowledge in a more extractable format (e.g., asking questions) after knowledge injection. Furthermore, Jiang et al. [19] has shown that the instruction-tuned model is better at learning new knowledge. Compared to previous works, we focus on developing an alternative method to data augmentation that perturbs the latent representation of LLMs for better knowledge injection.

**Data Augmentation and Latent Perturbation** The usefulness of data augmentations for text data was empirically observed in the literature. For instance, EDA [52] has introduced simple data augmentation method which randomly deletes, swaps, replaces, and inserts the words. Other previous works [22, 5, 30] have utilized the trained LMs to augment the text data. Recently, Maini et al. [29] has shown that adding data rephrased by LLMs into the pre-training corpus improves the performance of LM pre-training. However, those methods require additional costs in the knowledge injection as it utilize the LLMs to rephrase the text. In contrast, the latent perturbations offer an orthogonal approach to improve the robustness of neural networks, complementing data augmentation. This technique has been employed in meta-learning and out-of-distribution generalization [24, 25, 40]. For instance, NEFTune [16] demonstrated that adding noise, randomly sampled from a uniform distribution, to token embedding layers improves instruction tuning performance. Expanding on the concept of latent perturbations, our work introduces a novel approach that *internalizes* the effects of text paraphrasing by identifying optimal latent perturbations through training a small neural network within the LLMs.

## 3 Problem Formulation

In this work, we follow the knowledge injection setting outlined by Ovadia et al. [33]. We are given three resources: (1) documents $\mathcal{D}_\mathsf{K}$ containing knowledge that we are interested to inject; (2) question & answering dataset $\mathcal{D}_\mathsf{QA} = \{(\boldsymbol{q}^{(i)}, \boldsymbol{a}^{(i)})\}_{i=1}^{n}$ for verifying injected knowledge from $\mathcal{D}_\mathsf{K}$; and (3) a pre-trained Large Language Models (LLMs) $p_\theta(\cdot)$ parameterized by $\theta$. Our objective is to find a transformation $F$ that could enhance the knowledge about $\mathcal{D}_\mathsf{QA}$:

$$\theta' = F(\theta, \mathcal{D}_\mathsf{K}) \quad \text{s.t.} \quad \mathcal{S}(\theta', \mathcal{D}_\mathsf{QA}) > \mathcal{S}(\theta, \mathcal{D}_\mathsf{QA}), \tag{1}$$

where the score function $\mathcal{S}$ is defined as:

$$\mathcal{S}(\theta, \mathcal{D}_\mathsf{QA}) \coloneqq \frac{\sum_{i=1}^{n} \mathbb{I}(f(p_\theta(\boldsymbol{q}^{(i)})) = \boldsymbol{a}^{(i)})}{n}, \tag{2}$$

and $\mathbb{I}(\cdot)$ and $f(\cdot)$ denote the indicator function and a decoding function that samples a sequence of tokens from $p_\theta$, respectively.

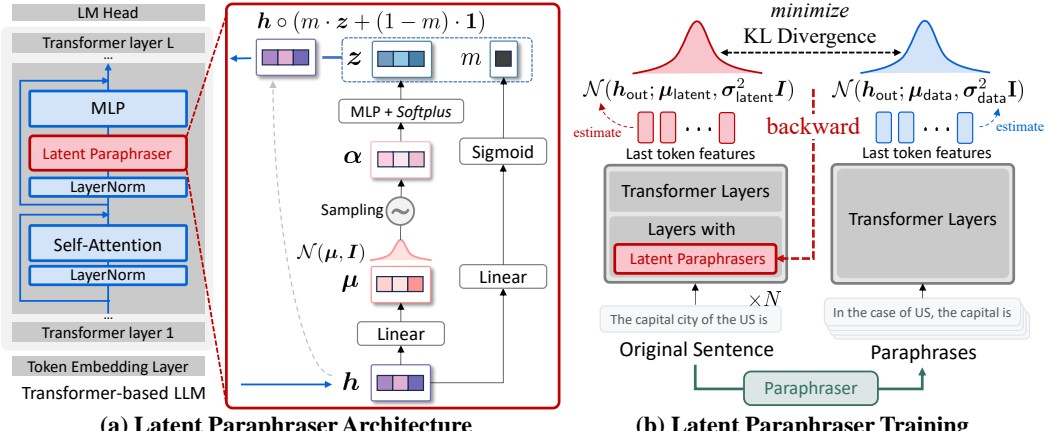

**(a) Latent Paraphraser Architecture**    **(b) Latent Paraphraser Training**

Figure 3: **(a) Illustration of the latent paraphraser.** The linear layer embeds each token's latent feature $h$ into $\mu$. We then sample stochastic noise $\alpha$ from $\mathcal{N}(\mu, I)$ and apply a mask $m_t$ to control the scale. **(b) Training pipeline of LaPael.** To train the latent paraphraser, we estimate the parameters of Gaussian distributions. We then minimize the KL divergence between these distributions to optimize the latent paraphrasers.

In general, a transformation $F$ is a fine-tuning LLMs on documents in $\mathcal{D}_K$ by optimizing $\theta$ to minimize the negative log-likelihood of each token in each document as follows [33]:

$$\theta^* = \arg\min_\theta \frac{1}{|\mathcal{D}_K|} \sum_{s \in \mathcal{D}_k} \left( \frac{1}{|s|} \sum_{t=1}^{|s|} -\log p_\theta(s_t \mid s_{<t}) \right), \tag{3}$$

where $|s|$ denotes the length of token sequence $s$.

## 4  Proposed Method

We propose Latent Paraphrasing of Language Models (LaPael), a framework that perturbs the latent feature of LLMs, to achieve the equivalent effect of data augmentation at the latent level. Knowledge injection using LaP consists of the following four processes: paraphrasing the set of documents to make the paraphrased data (Section 4.1), training the latent paraphrasers with paraphrased data (Section 4.2), fine-tuning LLMs with the trained latent paraphrasers on $\mathcal{D}_K$ and evaluate the injected knowledge of LLMs on $\mathcal{D}_{QA}$ (Section 4.3).

### 4.1  Data Augmentation: Paraphrasing

To train the latent paraphrasers, we need a distinct set of training data $\mathcal{D}_{\text{train}} = \{s^{(i)}\}_{i=1}^N$ which consists of documents having different knowledge with $\mathcal{D}_K$. As a preliminary, we formulate the paraphrasing of the text in terms of the **knowledge equivalence**, which is a narrower concept than semantic equivalence [23] where two different sentences can contain the same knowledge. We consider that each sentence $s$ in $\mathcal{D}_{\text{train}}$ can be decomposed into words for the object (entity or attribute) of the knowledge ($y$) and others ($x$) where both are the sequence of tokens. For instance, given the sentence *"The capital of the United States is Washington D.C."*,

$$x = \textit{"The capital of the United States is"}; \quad y = \textit{"Washington D.C."},$$

represent the knowledge (United States, capital, Washington D.C.). Then, we paraphrase a sentence $s = (x, y)$ into a paraphrased sentence[1]. For the above sentence, a paraphrased sentence can be

$$x' = \textit{"In the case of the United States, the designated capital city is"}$$

with the same $y$, which is knowledge equivalent to $(x, y)$. For each knowledge $\mathcal{K}$, we assume that there is a set of the knowledge equivalent sentences $S(\mathcal{K})$ where $(x, y) \in S(\mathcal{K})$. We generate $K$ paraphrased sentence via a LLM: $(x_1, y), \dots, (x_K, y) \sim p_{\text{LLM}}(x' | \text{prompt}, x, y)$. Then, we have the set of paraphrased data $\{\{(x_k^{(i)}, y^{(i)})\}_{k=1}^K\}_{i=1}^N$ of $\mathcal{D}_{\text{train}}$. We define $p(x'|x) := p_{\text{LLM}}(x' | \text{prompt}, x, y)$ which denotes the probability distribution of paraphrases given the original sentence.

---

[1]One possible way is to prompt the LLM (e.g., `gpt-3.5-turbo` [31]) with instruction *"For the following paragraph give me a paraphrase of the same in high-quality English language as in sentences on Wikipedia"* [29]

## 4.2 Introducing Latent Paraphraser

**Latent Paraphraser**   We introduce a latent paraphraser within a transformer layer [50], which augments a latent feature and is expected to paraphrase the given input text within the latent space. As illustrated in Figure 3(a), within the transformer architecture, we insert this new layer just before the Multi-layer Perceptron (MLP), using the output from the second LayerNorm as its input.

Let $\boldsymbol{h} \in \mathbb{R}^d$ denote the latent feature after the second LayerNorm. The latent paraphraser, denoted by $g_\phi : \mathbb{R}^d \to \mathbb{R}^d$ and parameterized by $\phi$, augments the latent feature as follows:

$$\boldsymbol{h} \circ g_\phi(\boldsymbol{h}), \tag{4}$$

where $\circ$ is the element-wise multiplication. The function $g_\phi(\boldsymbol{h})$ is given by:

$$g_\phi(\boldsymbol{h}) = (1 - m) \cdot \mathbf{1} + m \cdot \boldsymbol{z}, \tag{5}$$

with $\boldsymbol{z} \in \mathbb{R}^d$ and $m \in [0, 1]$ representing a noise vector and a learnable mask, respectively.

The noise vector $\boldsymbol{z}$ is generated by

$$\boldsymbol{z} = \mathrm{softplus}(\mathrm{MLP}_{\boldsymbol{z}}(\boldsymbol{\alpha})), \quad \boldsymbol{\alpha} \sim \mathcal{N}(\boldsymbol{\mu}, \boldsymbol{I}), \quad \boldsymbol{\mu} = \boldsymbol{W}_\mu \boldsymbol{h} + \boldsymbol{b}_\mu, \tag{6}$$

where $\mathrm{MLP}_{\boldsymbol{z}}$ is a 2-layers MLP. We use the reparameterization trick [21] to enable the back-propagation through the sampling from the Gaussian distribution: $\boldsymbol{\alpha} = \boldsymbol{\mu} + \boldsymbol{\epsilon}$, where $\boldsymbol{\epsilon} \sim \mathcal{N}(0, \boldsymbol{I})$.

To modulate the scale of perturbation for individual tokens, we employ a learnable mask. It is important as too much noise on key tokens (e.g., United States) might hurt the semantics of the sequence. For learnable binary mask, we use concrete distribution to approximate the sampling discrete random variable from a Bernoulli distribution using continuous relaxation [8] as follows:

$$m = \mathrm{sigmoid}\left(\frac{1}{\tau} \log(u) + \log(1 - u) + \tilde{m}\right), \quad \tilde{m} = \boldsymbol{W}_m \boldsymbol{h} + b_m, \tag{7}$$

where $u \sim \mathrm{Unif}(0, 1)$, $\tau$ is temperature, and $m$ is mask value in scalar.

**Training**   Then, how do we train the latent paraphrasers to approximate optimal perturbation functions for estimating the distribution of the paraphrased text? We employ the dataset with paraphrases $\{\{(\boldsymbol{x}_k^{(i)}, \boldsymbol{y}^{(i)})\}_{k=1}^K\}_{i=1}^N$ generated in Section 4.1. Our objective is to match two distributions for each transformer layer:

1. the distribution of transformer layer output feature for the last token $\boldsymbol{h}_{\mathrm{out}}$ without the latent paraphraser given the data perturbation distribution $p(\boldsymbol{x}'|\boldsymbol{x})$ from Section 4.1:

$$p_\theta(\boldsymbol{h}_{\mathrm{out}}|\boldsymbol{x}) = \int p_\theta(\boldsymbol{h}_{\mathrm{out}}|\boldsymbol{x}')p(\boldsymbol{x}'|\boldsymbol{x})d\boldsymbol{x}'; \tag{8}$$

2. the distribution of output feature for the last token $\boldsymbol{h}_{\mathrm{out}}$ with the latent paraphraser given $\boldsymbol{x}$, $p_{\theta,\phi}(\boldsymbol{h}_{\mathrm{out}}|\boldsymbol{x})$. As a latent paraphraser outputs stochastic noise, we can formulate the probabilistic distribution $p_{\theta,\phi}(\boldsymbol{h}_{\mathrm{out}}|\boldsymbol{x})$ as follows:

$$p_{\theta,\phi}(\boldsymbol{h}_{\mathrm{out}}|\boldsymbol{x}) = \int p_\theta(\boldsymbol{h}_{\mathrm{out}} \mid \boldsymbol{x}, \boldsymbol{z})p_{\theta,\phi}(\boldsymbol{z} \mid \boldsymbol{x})d\boldsymbol{z}, \tag{9}$$

where $p_{\theta,\phi}(\boldsymbol{z} \mid \boldsymbol{x})$ is the distribution for noise from the latent paraphraser in Equation (6).

We make the simplistic parametric assumption that both distributions are Gaussian:

$$p_\theta(\boldsymbol{h}_{\mathrm{out}}|\boldsymbol{x}) \sim \mathcal{N}(\boldsymbol{h}_{\mathrm{out}}; \boldsymbol{\mu}_{\mathrm{data}}, \boldsymbol{\sigma}_{\mathrm{data}}^2 \boldsymbol{I}); \quad p_{\theta,\phi}(\boldsymbol{h}_{\mathrm{out}}|\boldsymbol{x}) \sim \mathcal{N}(\boldsymbol{h}_{\mathrm{out}}; \boldsymbol{\mu}_{\mathrm{latent}}, \boldsymbol{\sigma}_{\mathrm{latent}}^2 \boldsymbol{I}). \tag{10}$$

To train latent paraphrasers, we minimize the symmetric Kullback-Leibler (KL) divergence between two estimated Gaussian distributions of each layer as follows:

$$\mathcal{L}_{\mathsf{KL}}(\boldsymbol{x}) = \frac{1}{2}(\hat{D}_{\mathsf{KL}}(p_\theta(\boldsymbol{h}_{\mathrm{out}}|\boldsymbol{x})\|p_{\theta,\phi}(\boldsymbol{h}_{\mathrm{out}}|\boldsymbol{x})) + \hat{D}_{\mathsf{KL}}(p_{\theta,\phi}(\boldsymbol{h}_{\mathrm{out}}|\boldsymbol{x})\|p_\theta(\boldsymbol{h}_{\mathrm{out}}|\boldsymbol{x}))), \tag{11}$$

$$\hat{D}_{\mathsf{KL}}(p_\theta(\boldsymbol{h}_{\mathrm{out}}|\boldsymbol{x})\|p_{\theta,\phi}(\boldsymbol{h}_{\mathrm{out}}|\boldsymbol{x})) = \log\left(\frac{\hat{\boldsymbol{\sigma}}_{\mathrm{latent}}}{\hat{\boldsymbol{\sigma}}_{\mathrm{data}}}\right) + \frac{\hat{\boldsymbol{\sigma}}_{\mathrm{data}}^2 + (\hat{\boldsymbol{\mu}}_{\mathrm{data}} - \hat{\boldsymbol{\mu}}_{\mathrm{latent}})^2}{2\hat{\boldsymbol{\sigma}}_{\mathrm{latent}}^2} - \frac{1}{2}. \tag{12}$$

We employ a Monte Carlo sampling approach to estimate the parameters of Gaussian distributions. We generate $N$ samples $\boldsymbol{h}_{\mathsf{latent}}^{(1)}, \ldots, \boldsymbol{h}_{\mathsf{latent}}^{(N)}$ from the distribution $p_{\theta,\phi}(\boldsymbol{h}_{\mathsf{out}} \mid \boldsymbol{x})$. Then, we estimate the empirical mean and standard deviation from the samples as follows:

$$\hat{\boldsymbol{\mu}}_{\mathsf{latent}} = \frac{1}{N} \sum_{i=1}^{N} \boldsymbol{h}_{\mathsf{latent}}^{(i)}, \quad \hat{\boldsymbol{\sigma}}_{\mathsf{latent}} = \sqrt{\frac{1}{N-1} \sum_{i=1}^{N} (\boldsymbol{h}_{\mathsf{latent}}^{(i)} - \hat{\boldsymbol{\mu}}_{\mathsf{latent}})^2}, \tag{13}$$

and we use $K$ paraphrases $\boldsymbol{x}_1, \ldots, \boldsymbol{x}_K$ to obtain $K$ samples $\boldsymbol{h}_{\mathsf{data}}^{(1)}, \ldots, \boldsymbol{h}_{\mathsf{data}}^{(K)}$ from the distribution $p_\theta(\boldsymbol{h}_{\mathsf{out}} \mid \boldsymbol{x})$. Then we estimate the parameters in the same way:

$$\hat{\boldsymbol{\mu}}_{\mathsf{data}} = \frac{1}{K} \sum_{k=1}^{K} \boldsymbol{h}_{\mathsf{data}}^{(k)}, \quad \hat{\boldsymbol{\sigma}}_{\mathsf{data}} = \sqrt{\frac{1}{K-1} \sum_{k=1}^{K} (\boldsymbol{h}_{\mathsf{data}}^{(k)} - \hat{\boldsymbol{\mu}}_{\mathsf{data}})^2}. \tag{14}$$

We further use the auxiliary loss for mask training, with the sequence length of $T$ as follows:

$$\mathcal{L}_{\mathsf{mask}}(\boldsymbol{x}) = \sum_{t=1}^{T} \left( |\mathrm{sigmoid}(\tilde{m}_t) - r \cdot T| + |\mathrm{sigmoid}(\tilde{m}_t) - \bar{m}_t| \right), \tag{15}$$

where $\tilde{m}_T$ is defined in Equation (7), $r \in [0, 1]$ is the mask ratio that controls the number of masks and $\bar{m}_t$ is the gold mask where $\bar{m}_t = 0$ for tokens that correspond to the named entity.

To sum up, we optimize the latent paraphraser parameter $\phi$ by minimizing the following loss:

$$\phi^* = \arg\min_{\phi} \sum\nolimits_{\boldsymbol{x} \in \mathcal{D}_{\mathsf{train}}} \left( \mathcal{L}_{\mathsf{KL}}(\boldsymbol{x}) + \mathcal{L}_{\mathsf{mask}}(\boldsymbol{x}) \right). \tag{16}$$

See Figure 3(b) for an illustration of the training process for the latent paraphraser.

## 4.3 Fine-tuning the LLM with the Trained Latent Paraphrasers

We fine-tune the LLM on documents containing knowledge to be injected ($\mathcal{D}_{\mathsf{K}}$) as in Equation (3). We use the trained latent paraphraser parameterized by $\phi^*$ during LLM fine-tuning as follows:

$$\theta^* = \arg\min_{\theta} \frac{1}{|\mathcal{D}_{\mathsf{k}}|} \sum_{\boldsymbol{s} \in \mathcal{D}_{\mathsf{k}}} \left( \frac{1}{|\boldsymbol{s}|} \sum_{t=1}^{|\boldsymbol{s}|} \left( \frac{1}{N} \sum_{j=1}^{N} -\log p_{\theta,\phi^*}(\boldsymbol{s}_t \mid \boldsymbol{z}_t^{(j)}, \boldsymbol{s}_{<t}) p_{\theta,\phi^*}(\boldsymbol{z}_t^{(j)} \mid \boldsymbol{s}_{<t}) \right) \right), \tag{17}$$

where we sample $N$ noise $\boldsymbol{z}^{(j)}$ by sampling multiple $\boldsymbol{\alpha}$ from Gaussian distribution as defined in Equation (6). Then, we evaluate the knowledge injected in LLMs by measuring $\mathcal{S}(\theta^*, \mathcal{D}_{\mathsf{QA}})$ as defined in Equation (2).

# 5 Experiments

In experiments, we validate the effectiveness of the proposed method, LaPael, in injecting new or under-represented knowledge into Large Language Models (LLMs).

## 5.1 Experimental Setting

### 5.1.1 Datasets

To follow the experimental setup in Section 3, we need (1) documents containing knowledge $\mathcal{D}_{\mathsf{K}}$ and (2) associated QA datasets $\mathcal{D}_{\mathsf{QA}}$. We mainly use the test split of three QA datasets: SQuAD [38], StreamingQA [27], and ArchivalQA [51] for the source of $\mathcal{D}_{\mathsf{K}}$ and $\mathcal{D}_{\mathsf{QA}}$ in our main experiments. These datasets, previously used in Hu et al. [14], consist of documents paired with their corresponding QAs, making them well-suited to our experimental setup. While the questions in these datasets are of decent quality, a significant limitation lies in the documents provided. These documents are likely to have been seen by LLMs during pre-training, making it difficult to accurately assess the performance of methods on injecting new knowledge.

Table 1: **Data Example**. Example data from SQuAD and StreamingQA dataset we used in experiments. Words in the yellow background indicate the answer to the question. More examples are in Table 12 of the Appendix.

| Question | Raw Document | Synthetic Document |
|---|---|---|
| **Who was the Super Bowl 50 MVP?** *(from SQuAD)* | (...) Denver linebacker Von Miller was named Super Bowl MVP, recording five solo tackles, 2 sacks, and two forced fumbles. | The Super Bowl 50 MVP was Von Miller. |
| **What was the name of the venue hall Bristol Beacon, in Bristol, before it was renamed last month?** *(from StreamingQA)* | (...) Colston Hall, which was named after the 17th-century slave trader Edward Colston, will from now on be known as Bristol Beacon following a public consultation. Bristol attracted headlines around (...) | Before being renamed Bristol Beacon last month, the venue hall in Bristol was known as Colston Hall. |

Table 2: Experimental results on datasets with **synthetic documents**. *trained with n sents* means that latent paraphrasers are trained with the dataset containing $n$ sentences. For ours, we report the average performance of three runs. † denotes the method that uses 10 times more additional data (paraphrases).

| Method | SQuAD-syn | | | StreamingQA-syn | | | ArchivalQA-syn | | |
|---|---|---|---|---|---|---|---|---|---|
| | EM | Recall | F1 | EM | Recall | F1 | EM | Recall | F1 |
| **No Injection** | 13.10 | 22.91 | 21.09 | 16.39 | 26.30 | 23.71 | 13.50 | 25.07 | 22.12 |
| **Fine-Tuning** | 66.30 | 79.32 | 76.11 | 82.08 | 88.98 | 88.29 | 62.60 | 79.51 | 76.16 |
| **Fine-Tuning** *(seq.)* | 67.60 | 80.30 | 77.39 | 77.95 | 86.36 | 85.23 | 56.30 | 79.17 | 74.12 |
| **FreeLB** [57] | 70.70 | 82.41 | 79.67 | 82.24 | 89.48 | 88.56 | 63.20 | 81.30 | 77.67 |
| **NEFTune** [16] | 68.30 | 80.93 | 77.91 | 81.47 | 88.66 | 87.77 | 61.90 | 78.90 | 75.81 |
| **Ours** *trained w/ 50 sents.* | 70.77 | 84.96 | 81.66 | **86.16** | **93.01** | **92.12** | 68.37 | 86.24 | 82.67 |
| **Ours** *trained w/ 1k sents.* | **72.47** | **87.93** | **84.50** | 84.48 | 92.42 | 91.33 | **68.37** | **88.99** | **84.75** |
| **Fine-Tuning** *(+ para.)*† | 68.50 | 85.12 | 80.51 | 85.45 | **93.67** | **92.32** | 64.90 | 85.92 | 81.24 |

To mitigate this issue, we incorporate two datasets with synthetic QAs – Films 2024 and Events 2024. These are QA datasets generated from raw Wikipedia articles under the 2024 films category and from US events in May, June, and July 2024, in the 2024 events in the United States category. We generated question-answer pairs from these documents using GPT-4o following methods from previous works [19, 33]. Since the documents used to generate these datasets were not seen by the LLMs during pre-training, we can better evaluate the effectiveness of each method for knowledge injection especially on new knowledge.

**Datasets with Synthetic Documents**    The raw documents from datasets are unsuitable for precisely measuring the knowledge injection performance. Specifically, fine-tuning LLMs on a document does not always ensure that LLMs can answer the associated questions, due to the reversal curse [3]. Moreover, documents often contain irrelevant knowledge that may hinder the accurate assessment of knowledge injection [14].

To address these issues, we conduct evaluations under the setting of synthetic documents. For generating synthetic documents, we construct $\mathcal{D}_K$ by rephrasing each question and answer in $\mathcal{D}_{QA}$ using GPT-4-turbo [32], ensuring that fine-tuning on these synthetic documents guarantee that LLMs become answerable to the associated questions. Examples of questions, synthetic, and raw documents are shown in Table 1. To make a difference, we denote the dataset under the synthetic document setting with the suffix '-syn' and the raw document setting with the suffix '-raw'.

**Datasets for Training Latent Paraphrasers**    For training our latent paraphrasers, the set of training data $\mathcal{D}_{train}$ is required in addition to $\mathcal{D}_K$. Therefore, we use GPT-3.5-turbo [31] to generate the set of synthetic sentences from the subset of a training split of each QA dataset, where each sentence must be with the answer to questions, following the sentence format in Section 4.1.

### 5.1.2 Experimental Details

**Baselines**    We compare our LaPael against several baselines. All models are fine-tuned on the documents in $\mathcal{D}_K$ unless explicitly stated otherwise. **(1) No Injection.** We use the pre-trained LLM without any fine-tuning. **(2) Fine-Tuning.** We fine-tune the LLM on $\mathcal{D}_K$. **(3) Fine-Tuning** *(seq)*. We first fine-tune the LLM on the paraphrased documents of $\mathcal{D}_{train}$. Then, we fine-tune the LLM on $\mathcal{D}_K$. **(4) Fine-Tuning** *(+ para)*. We fine-tune LLM on the original and paraphrased documents of $\mathcal{D}_K$. **(5) FreeLB** [57]. We add trained adversarial noise to the token embedding while fine-tuning. **(6) NEFTune**[16]. We add random uniform noise to the token embedding while fine-tuning. **(7) LaPael (ours).** We train the latent paraphrasers on $\mathcal{D}_{train}$ and then fine-tune the model on $\mathcal{D}_K$.

Table 3: Experimental results on datasets with **raw documents**. For Ours, we use the latent paraphraser used in the SQuAD-syn experiment. Rec. denotes recall.

| Method | SQuAD-raw | | | StreamingQA-raw | | | Films 2024-raw | | | Events 2024-raw | | |
|---|---|---|---|---|---|---|---|---|---|---|---|---|
| | EM | Rec. | F1 | EM | Rec. | F1 | EM | Rec. | F1 | EM | Rec. | F1 |
| No Injection | 9.98 | 23.44 | 20.62 | 16.22 | 29.85 | 27.81 | 1.93 | 10.21 | 10.27 | 1.73 | 17.94 | 17.40 |
| Fine-Tuning | 16.65 | 35.40 | 29.73 | 19.04 | 35.88 | 32.92 | 13.39 | 30.03 | 28.84 | 10.98 | 43.76 | 39.62 |
| FreeLB [57] | 17.04 | 36.78 | 30.36 | 20.72 | 37.19 | 34.04 | 15.47 | 33.69 | 31.85 | 14.68 | 46.05 | 41.85 |
| NEFTune [16] | 17.45 | 37.49 | 31.11 | 20.18 | 36.98 | 33.85 | 15.93 | 33.73 | 32.38 | **15.38** | 48.14 | 43.84 |
| **Ours** | **18.96** | **43.10** | **34.65** | **21.62** | **39.38** | **35.32** | **16.29** | **35.04** | **32.56** | 15.26 | **56.70** | **46.45** |

Table 4: Experimental results on **cross-domain transfer** experiments. For ours, (X →) denotes that latent paraphrasers are trained on $\mathcal{D}_{\text{train}}$ from the X dataset. Rec. denotes recall.

| Method | SQuAD-syn | | | StreamingQA-syn | | | NovelQA-syn | | | MedMCQA-syn | | |
|---|---|---|---|---|---|---|---|---|---|---|---|---|
| | EM | Rec. | F1 | EM | Rec. | F1 | EM | Rec. | F1 | EM | Rec. | F1 |
| No Injection | 13.10 | 22.91 | 21.09 | 16.39 | 26.30 | 23.71 | 9.17 | 18.21 | 16.05 | 39.00 | 48.68 | 47.82 |
| Fine-Tuning | 58.30 | 68.59 | 66.35 | 74.73 | 82.34 | 81.21 | 52.92 | 66.30 | 63.62 | 56.10 | 62.37 | 62.03 |
| FreeLB [57] | 70.70 | 82.41 | 79.67 | 82.24 | 89.48 | 88.56 | **55.42** | 67.39 | 64.80 | 57.90 | 63.17 | 62.81 |
| NEFTune [16] | 68.30 | 80.93 | 77.91 | 81.47 | 88.66 | 87.77 | 51.67 | 65.14 | 62.25 | 56.30 | 62.57 | 62.09 |
| **Ours (SQuAD →)** | 72.50 | 89.38 | 85.34 | **84.38** | 93.44 | **92.17** | 54.17 | 69.40 | 65.72 | **63.70** | **68.28** | **67.98** |
| **Ours (StreamingQA →)** | **72.80** | **89.65** | **85.90** | 84.06 | 93.73 | 91.90 | 54.58 | **72.58** | **68.15** | 63.20 | 68.02 | 67.79 |

**Training & Inference**   We mainly use Vicuna-7b-v1.5 [56] for fine-tuning, which is the instruction-tuned version of Llama-2-7b [48] for our experiments. We fine-tune LLMs for 12 epochs with a learning rate of 0.00005 and step learning rate scheduler where we decay a learning rate by 0.85 by every 4 epochs. For inference, we use in-context learning with 5 examples by prompting the 5 examples in the prompt [4]. To measure QA accuracy, we use Exact Match (EM), Recall (Rec.), and F1 score. More details on the experimental setting are provided in the Appendix C.

## 5.2   Experimental Results

**Experiments with Synthetic Documents**   In Table 2, we present the experimental results for the synthetic documents setting. Fine-tuning does improve the QA performance of LLMs, but it does not lead to near-perfect scores even though the synthetic document contains the necessary knowledge for answering the questions, as shown in Table 1.

Our experiments show that paraphrasing documents for fine-tuning consistently improves QA performance across all three benchmarks. Notably, LaPael demonstrates performance comparable to fine-tuning with paraphrases on StreamingQA and even outperforms it on two other benchmarks. These findings suggest that the latent paraphrasers learn an effective noise distribution that aids knowledge injection without additional data augmentation.

We also compared LaPael with two other noise-based methods, FreeLB [57] and NEFTune [16], to validate that the latent-level noise generated by latent paraphrasers is more effective. As shown in Table 2, LaPael outperforms these baselines, confirming the strength of our approach.

**Experiments with Raw Documents**   While our method has proven effective for knowledge injection with synthetic documents, it is important to evaluate its performance on raw documents, which represent a more realistic data format. To demonstrate the applicability of our method to real-world data, we conducted experiments in which we fine-tuned LLMs on raw documents for each dataset, using latent paraphrasers trained on $\mathcal{D}_{\text{train}}$ from SQuAD-syn.

As shown in Table 3, our method outperforms both fine-tuning and noise-based baselines in the context of knowledge injection with raw documents. Considering that the latent paraphrasers were trained on synthetic sentences from $\mathcal{D}_{\text{train}}$, these results demonstrate their effectiveness on documents with a different format than those used in training.

**Cross-domain Transfer**   Once trained, the latent paraphrasers can be applied to fine-tune LLMs on documents from any domain. To demonstrate this, we conducted cross-domain transfer experiments. Specifically, we trained latent paraphrasers on $\mathcal{D}_{\text{train}}$ from a source domain (e.g., SQuAD) and fine-tuned LLMs with the trained latent paraphrasers on $\mathcal{D}_{\text{K}}$ from a target domain (e.g., StreamingQA).

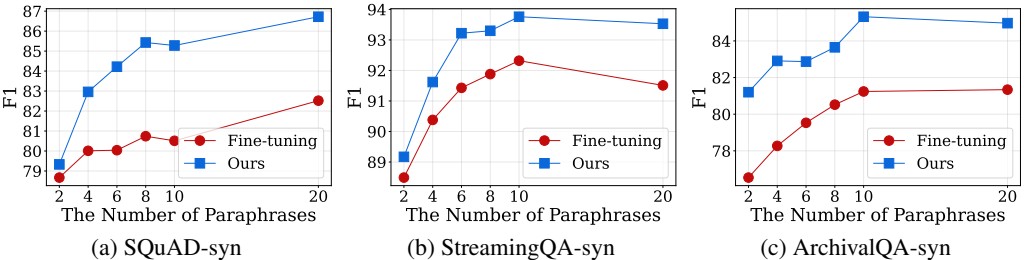

Figure 4: **Effect of the Number of Paraphrases.** Each plot shows the relationship between the number of paraphrases (x-axis) and F1 scores (y-axis) in knowledge injection. The F1 scores of both standard fine-tuning and our method improve as the number of paraphrases increases.

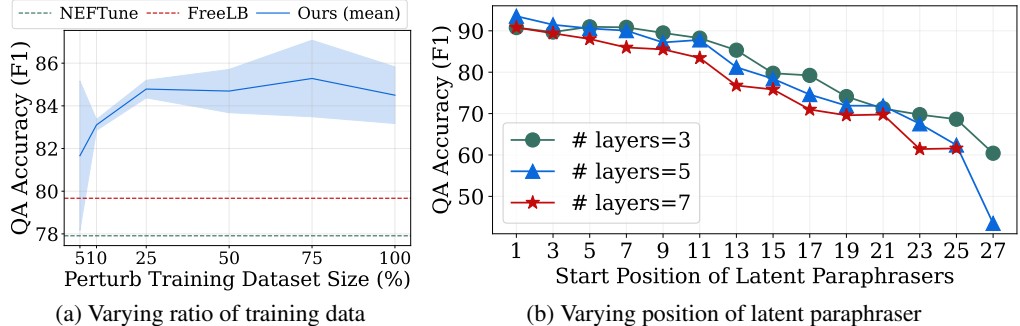

Figure 5: **(a)** We conduct experiments varying the size of $\mathcal{D}_{\text{train}}$ on SQuAD-syn, where $100\%$ indicates 1,000 documents. We report mean and std. over three runs. **(b)** We conduct experiments on StreamingQA-syn varying the start position of latent paraphrasers where '# layers' denotes the number of latent paraphrasers.

As shown in Table 4, our method successfully transfers across domains, with the latent paraphrasers enhancing the performance of the knowledge injection on NovelQA and MedMCQA–two domains distinct from the source (see Appendix C.1 for details on these datasets). Even though both domains contain specialized entities, our method consistently outperforms standard fine-tuning and other noise-based baselines.

**Combining LaPael and Paraphrases**    Paraphrasing documents in $\mathcal{D}_K$ has been shown to improve knowledge injection performance, as seen in Table 2. While LaPael significantly improves performance without requiring paraphrases, it is valuable to consider the effect of combining paraphrases with the latent perturbations from LaPael. As illustrated in Figure 4, LaPael consistently outperforms standard fine-tuning, showing that LaPael provides advantages over data-level augmentations.

## 5.3 Ablation Studies

**Effects of the Size of $\mathcal{D}_{\text{train}}$**    LaPael needs additional data $\mathcal{D}_{\text{train}}$ for training latent paraphrasers. Although only a small amount of data is required, it might be unclear how much is needed to make the latent paraphrasers learn the useful noise distribution. As shown in Figure 5a, LaPael works well even with **50 sentences** for $\mathcal{D}_{\text{train}}$, while increasing the size of $\mathcal{D}_{\text{train}}$ ensures a steady performance improvement for LaPael.

**Effects of the Position of Latent Paraphrasers**    Our latent paraphrasers can be inserted into any layer of the LLMs. The possible question is which position and how many layers are optimal for latent paraphrasers to effectively learn noise for knowledge injection. To answer this, we analyzed the position and number of latent paraphrasers.

In Figure 5b, we show the QA accuracy results, varying the start position and number of latent paraphrasers. The first layer is the closest layer to the input layer, and "start position 1" with "# layers = 3" means we insert the latent paraphrasers into the first, second, and third layers of the LLM. Results show that inserting three latent paraphrasers into the early layers of the LLM is effective. This is consistent with findings in previous works [16, 57, 25] where using noisy token embeddings (the lowest layer) enhanced the generalization in LLMs. Furthermore, in Table 5, we empirically show that positioning the latent paraphraser before the MLP layer within each transformer layer is the most effective choice over other positions.

Table 5: Analysis on the **Position** inside the Transformer layer.

| StreamingQA | EM | Recall | F1 |
|---|---|---|---|
| Before **MLP** | 84.06 | **93.73** | **91.90** |
| After **MLP** | 73.81 | 82.58 | 81.02 |
| Before **Attn** | 80.55 | 87.58 | 86.49 |
| After **Attn** | 83.31 | 90.98 | 89.72 |
| **Token Embed.** | **86.21** | 91.79 | 91.05 |

Table 6: Ablation studies on **Modules** in latent paraphrasers.

| StreamingQA | EM | Recall | F1 |
|---|---|---|---|
| **LaPael** | 84.06 | **93.73** | **91.90** |
| w/o **Mask** | 77.95 | 85.17 | 84.63 |
| w/o **Concrete** | 73.35 | 83.42 | 81.99 |
| w/o **Sampling** | **84.23** | 90.52 | 89.73 |
| w/o **KL loss** | 83.31 | 90.78 | 89.99 |

Table 7: Ablation studies on **Noise design** in latent paraphrasers.

| StreamingQA | EM | Recall | F1 |
|---|---|---|---|
| **Learnable Mul.** | **84.06** | **93.73** | **91.90** |
| **Learnable Add.** | 73.05 | 83.23 | 81.70 |
| **Gaussian** | 83.46 | 90.32 | 89.54 |
| **Gaussian** + *mask* | 82.85 | 89.70 | 88.87 |
| **Uniform** | 79.48 | 87.17 | 86.15 |
| **Uniform** + *mask* | 74.43 | 81.09 | 80.26 |

**Ablation Studies on Modules**    LaPael has many design choices concerning the latent paraphraser architecture, noise type, and training. We conducted extensive ablation studies to empirically verify each design choice and provide guidance for future work. In summary, as shown in Table 6, all design choices are important for building the most effective latent paraphraser. Specifically, we use a trainable mask $m$ in Equation (7) to regulate the perturbation depending on each token, which is crucial, as the performance on StreamingQA drops significantly if we remove it from the latent paraphraser. Furthermore, using only the sigmoid function in Equation (7) instead of the concrete distribution also leads to much lower performance, as the mask is not properly trained. Regarding noise training, using deterministic noise instead of stochastic noise by removing the noise drawn from a Gaussian distribution in Equation (5) also decreases performance. Additionally, replacing the KL loss with Mean Squared Error loss between two means $\hat{\boldsymbol{\mu}}_{\text{latent}}$ in Equation (13) and $\hat{\boldsymbol{\mu}}_{\text{data}}$ in Equation (14) leads to a decrease in performance, confirming the importance of stochastic noise trained with KL loss.

**Ablation Studies on Noise Distribution**    Should we train the latent paraphrasers to be effective, or can adding random noise in the early layers also be effective? Which is more important: the learnable mask or the learnable noise? To answer these questions, we conducted ablation studies on the choice of noise distribution. In Table 7, Learnable Add. denotes the model with the additive noise $\boldsymbol{h} + g_\phi(\boldsymbol{h})$ instead of Equation (4) without softplus from Equation (6). Gaussian is the use of zero-mean Gaussian noise $\mathcal{N}(\boldsymbol{0}, \boldsymbol{I})$ in Equation (6) without using $\text{MLP}_z$. Uniform is the use of noise drawn from the uniform distribution defined in NEFTune [16] instead of $z$ in Equation (6).

As shown in Table 7, the learnable multiplicative noise described in Section 4.2 is the best design for noise distribution used in the latent paraphraser. To analyze the effect of the learnable mask, we also added the learnable mask to the Gaussian and Uniform noise settings and optimized only $\boldsymbol{W}_m$ and $\boldsymbol{b}_m$ in Equation (7) with loss in Equation (15). Interestingly, the learnable mask is not effective for the fixed noise distribution, which contrasts with the results for learnable noise in Table 6. We conjecture that using the learnable mask is important for input-dependent learnable noise, as it can allocate different noise scales to different tokens, while this is not the case for static noise distribution.

# 6    Conclusion

We have introduced LaPael, a method for enhancing knowledge injection in Large Language Models (LLMs) by applying learned perturbations to their layers. Unlike traditional data-level augmentations or noise-based approaches, LaPael operates at the latent level, preserving the semantic integrity of the text while introducing meaningful variability. LaPael addresses key limitations of existing methods by reducing computational costs and increasing the diversity of augmented data. Our extensive validation across diverse benchmark datasets demonstrates the superiority of our method in knowledge injection, as it significantly outperforms both standard fine-tuning and other noise-based baselines. Moreover, combining LaPael with paraphrases yields complementary benefits, further enhancing performance. We believe that LaPael, being simple yet effective, has the potential for significant practical impact and will encourage further research on applying perturbation within the latent space of LLMs.

**Discussions & Limitations**    In our work, the following points can be discussed further: (1) Cost Analysis—While LaPael is effective, it incurs additional costs due to the need for training latent paraphrasers and fine-tuning LLMs with them. (2) Knowledge Retention—Although LaPael improves knowledge injection, there may be trade-offs in terms of retaining the original knowledge that the LLM has memorized. (3) Comparison to Retrieval-Augmented Generation (RAG)—While our method improves knowledge injection, it is still less effective than RAG in terms of performance. We provide a detailed discussion of these points, along with other limitations, in the Appendix A.

## Acknowledgement

We sincerely thank Byeongju Kim, Jongwon Jeong, Jimin Hong, and Jongho Park for their insightful discussion. This work was fully supported by the KRAFTON AI Research Center.

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

# Appendix

## A Discussions & Limitations

**Cost Analysis** Our method requires additional costs compared to the fine-tuning baseline. Specifically, it involves two extra computational costs beyond standard fine-tuning. A comparison of the per-step computational cost (in GFLOPs) between the baseline and our proposed method is shown in Table 8, where we consider fine-tuning LLMs with 7B parameters. In detail, one forward pass of a 7B parameter LLM requires 13.21 GFLOPs, and one backward pass costs twice as much as a forward pass. The latent paraphraser model we used in the experiments consists of 5 paraphrasers, each with 4 linear layers, totaling 250M parameters, which is 3.6% of the parameter size of the LLM. The total computational costs can vary depending on the hyperparameters (e.g., $N$ in Equation (13)) and the size of the dataset used.

While training the latent paraphrasers requires an initial cost, this is a one-time expense. Once trained, these can be used repeatedly for knowledge injection without additional ongoing costs. This makes the overall expense relatively low in the long term. Furthermore, incorporating latent paraphrasers during fine-tuning adds only a minimal computational overhead, as their parameter size is just 3.6% of the size of LLM.

**Knowledge Retention** A common drawback of knowledge injection is the potential for LLMs to forget previously learned knowledge [17]. To assess this issue, we used the EntityQuestions dataset [41], which contains simple questions about entities. Specifically, we focused on "place-of-birth" questions for well-known entities (e.g., "Where was Leonardo da Vinci born?"), with 988 questions in total. We fine-tune the Vicuna-7b-v1.5 [56] on a synthetic SQuAD document set ($\mathcal{D}_{\mathsf{K}}$) using each method, then measure its QA performance on the EntityQuestions.

As in Table 9, the experimental results show that all fine-tuning approaches negatively impact knowledge retention, as observed in the previous work [9]. Additionally, we observe that improved knowledge injection often comes at the cost of greater knowledge forgetting. Although our primary focus is on enhancing knowledge injection, we acknowledge that addressing knowledge retention is crucial and should be a focus of future research.

**Comparison to RAG** The primary advantages of fine-tuning methods, including ours, over retrieval-based approaches like Retrieval-Augmented Generation (RAG) [26], lie in their simplicity and reduced computational cost on the inference [6]. Fine-tuning results in a self-contained model, which simplifies the system architecture by removing the need for additional components like document retrieval and ranking during inference. This reduction in complexity leads to lower computational overhead, especially in terms of GPU memory usage due to the shorter length of the prompt, making fine-tuning more suitable for an LLM deployment in resource-constrained environments.

However, it is important to check the performance gap between them. Therefore, we experiment with RAG on the Events 2024 dataset with Vicuna-7b. For ours, we follow the same experimental setting with Table 3. For RAG, we use the bge-large-en-v1.5 [53] model for document and query embedding for retrieval. In Table 10, our experimental results indicate that the RAG approach outperforms fine-tuning methods including ours, as previously observed by de Luis Balaguer et al. [6]. However, our LaPael method narrows the gap between the two approaches, suggesting that there is potential for further improvements in fine-tuning strategies.

Table 8: Per-step computational cost comparison on the 7B LLM.

| Method | GFLOPs |
|---|---|
| **Baselines** | |
| Fine-tuning LLM | 39.63 |
| **Proposed Method** | |
| Training Latent Para. (LaP) | 14.64 |
| Fine-tuning LLM w/ LaP | 40.11 |

Table 9: Zero-shot question answering performance on EntityQuestions after fine-tuning LLMs on SQuAD-raw.

| | EM | Rec. | F1 |
|---|---|---|---|
| **No Injection** | **59.00** | **64.38** | **63.46** |
| **Fine-Tuning** | 52.23 | 55.63 | 55.41 |
| **Ours** | 39.88 | 41.97 | 42.12 |
| **Fine-Tuning** (+para) | 33.50 | 35.18 | 35.28 |

Table 10: Comparison to Retrieval Augmented Generation (RAG) on **Events 2024**-raw.

| | EM | Rec. | F1 |
|---|---|---|---|
| **Fine-Tuning** | 10.98 | 43.76 | 39.62 |
| **Ours** | 15.26 | 56.70 | 46.45 |
| **RAG** | **27.17** | **64.02** | **55.71** |

**Reversal curse.** The proposed method is unable to address the reversal curse, where the Large Language Models (LLMs) trained on "A is B" fail to answer "What is B?" [3]. As outlined in Berglund

et al. [3], this phenomenon is mainly due to the format of data and the autoregressive nature of LLMs that are trained in a way from left to right. Therefore, it is limited to improve the knowledge injection performance if the document does not contain a sentence having the reverse relationship, even with our method. Future work will need to explore the combining of our method with a recent solution for the reversal curse like reverse training [10]. Otherwise, we can seek a solution that addresses the reversal curse at the latent level similar to LaPael, which can be an interesting direction for future work.

**Limited scope of Task and Experiments.** The scope of our method remains limited in the knowledge injection task. Specifically, there are challenges in applying LaPael for continual pre-training on large-scale corpora, such as the 15B OpenWebMath dataset [35], or for instruction tuning with datasets like Alpaca [46]. Addressing these challenges will require future work as a new approach for training latent paraphrasers tailored to other tasks. In terms of experiments, our experiments only focus on the 7B LLMs, and do not conduct any experiment on larger LLMs of size with 13B or 70B [48] due to the limited computational budget for our experiments.

# B   Broader Impact

This work explores the knowledge injection in Large Language Models (LLMs), which are highly related to hallucinations [15]. While our method improves the addition of new knowledge to LLMs, it also increases the risk of introducing misinformation. Specifically, our method could enhance the inaccuracies in LLMs when they are fine-tuned using documents that contain incorrect facts. Therefore, it is crucial to thoroughly check the documents used for fine-tuning LLMs before applying our method to enhance knowledge injection.

# C   Experimental Details

## C.1   Dataset

Table 11: **Dataset statistics.** We report the size of $\mathcal{D}_{\text{train}}$, $\mathcal{D}_{\text{K}}$, and $\mathcal{D}_{\text{QA}}$ used in our experiments.

| Dataset | Synthetic Documents | | | | | Raw Documents | | | |
|---|---|---|---|---|---|---|---|---|---|
| | SQuAD | StreamingQA | ArchivalQA | NovelQA | MedMCQA | SQuAD | StreamingQA | Films 2024 | US Events 2024 |
| $\mathcal{D}_{\text{train}}$ | 1,000 | 1,000 | 1,000 | - | - | - | - | - | - |
| $\mathcal{D}_{\text{K}}$ | 1,000 | 653 | 1,000 | 240 | 1,000 | 2,067 | 1,628 | 1,202 | 175 |
| $\mathcal{D}_{\text{QA}}$ | 1,000 | 653 | 1,000 | 240 | 1,000 | 10,570 | 1,665 | 5,968 | 865 |

As briefly mentioned in Section 5.1, we generate the synthetic document from each question-answer pair using GPT-4-turbo model [32]. To generate the documents from the question and answer pairs, we use the prompt in Table 13. To generate diverse paraphrases from $\mathcal{D}_{\text{train}}$, we use the prompt [29] in Table 14 using GPT-3.5-turbo model. For cross-domain transfer experiments, we also use the subset of MedMCQA [34] and a synthetic NovelQA dataset based on the *Les Misérables* Wikipedia page, where we generate the synthetic document for each question. For MedMCQA [34], we use the subset of the dataset where the domain of question corresponds to the anatomy.

We summarize the statistics of the synthetic dataset used in our experiments in Table 11. We also plot the distributions of token counts in documents, questions, and answers for each dataset used in our experiments in Figure 6. We present the example of each dataset in Table 12.

## C.2   Training Details

As briefly mentioned in Section 5.1, we mainly use Vicuna-7b-v1.5 [56] for fine-tuning. We fine-tune LLMs for 12 epochs with a learning rate of 0.00005 and step learning rate scheduler where we decay a learning rate by 0.85 by every 4 epochs. For experiments in Figure 4, we fine-tune for 3 epochs with a decaying period as 1 epoch. For optimizer, we use AdamW [28]. For all experiments, we only update the parameters corresponding to the MLP layer of transformer [50]. For Llama model [47, 48], it corresponds to linear layers named `up_proj`, `gate_proj`, and `down_proj`. We use 4 A100 GPUs for fine-tuning LLMs. For inference, we use in-context learning with 5 examples by prompting the 5 examples in the prompt [4].

For training latent paraphrasers, we train them for 10 epochs with a learning rate of $1e-3$ and cosine learning rate scheduler where we linearly decay a learning rate to 10% of the initial learning rate without warmup. We use 5 latent paraphrasers on the 5 sequential early layers of LLMs. For Equation (13), we use $N = 4$. For Equation (14), we use $K = 10$. For Equation (15), we set $r = 0.5$. For gold mask $\bar{m}_t$, we use a similar method to Agrawal et al. [2] to find the named entities from each document using GPT-3.5-turbo. For fine-tuning with latent paraphrases (Equation (17)), we use $N = 4$.

Table 12: **Data Example**. Example data from all datasets we used in experiments. Words in the yellow background indicate the answer to the question. Hypen (-) in the original document column indicates the case where the original document is not accessible.

| Question | Original Document | Synthetic Document |
|---|---|---|
| **What is the name of Sudan's Prime Minister?** *(from StreamingQA)* | (...) In this Aug. 21, 2019 file photo, Sudan's new Prime Minister Abdalla Hamdok speaks during a press conference in Khartoum, Sudan. (...) | The Prime Minister of Sudan is Abdalla Hamdok. |
| **Which NFL team represented the NFC at Super Bowl 50?** *(from SQuAD)* | (...) The American Football Conference (AFC) champion Denver Broncos defeated the National Football Conference (NFC) champion Carolina Panthers to earn their third Super Bowl title. (...) | The NFC representative at Super Bowl 50 was the Carolina Panthers. |
| **What country's semi-official television network broadcast Bush's dinner?** *(from ArchivalQA)* | - | Bush's dinner was broadcast by the semi-official television network of Japan. |
| **Best graft for infra inguinal approach bypass (A) Dacron (B) PTFE (C) Polyester (D) Autologous vein** *(from MedMCQA)* | - | In infrainguinal bypass surgery, the preferred type of graft for optimal outcomes is an Autologous vein. |
| **What town does Jean Valjean become mayor of?** *(from NovelQA)* | - | Jean Valjean becomes the mayor of the town Montreuil-sur-Mer. |
| **How many titles were screened in person at the 23rd New York Asian Film Festival?** *(from Events 2024)* | The 23rd New York Asian Film Festival was held in New York on 12 July with World Premiere of South Korean film Victory by Park Beom-su, who attended the screening in person. In the 23rd edition 94 titles were screened in person. (...) | - |
| **Who directed and produced Dune: Part Two?** *(from Films 2024)* | Dune: Part Two is a 2024 epic science fiction film directed and produced by Denis Villeneuve, who co-wrote the screenplay with Jon Spaihts. The sequel (...) | - |

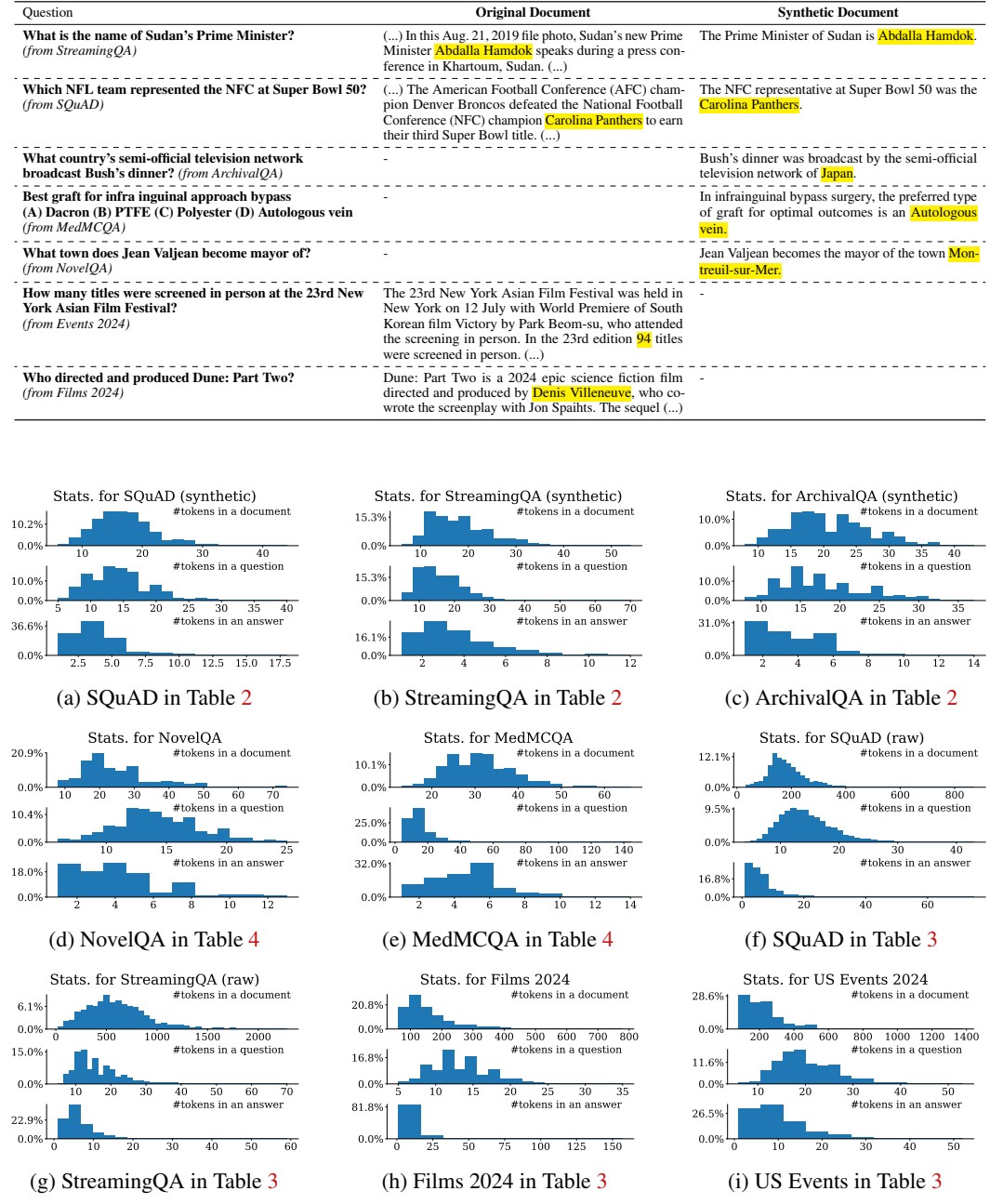

(a) SQuAD in Table 2  (b) StreamingQA in Table 2  (c) ArchivalQA in Table 2

(d) NovelQA in Table 4  (e) MedMCQA in Table 4  (f) SQuAD in Table 3

(g) StreamingQA in Table 3  (h) Films 2024 in Table 3  (i) US Events in Table 3

Figure 6: The distributions of token counts in documents, questions, and answers for each dataset used in our experiments.

Table 13: **Prompt for Synthetic Document Generation.** An 1-shot prompt for generating the synthetic document from the question.

Write a concise informative background sentence, that is directly helpful to answer the following question. The background sentence is the sentence that ends with a suffix. In other words, the answer entity should be followed by the entities used in the question.

### Question
Question: Who replaced Tim Sloan as CEO of Wells Fargo? Answer: Charles Scharf
### Suffix
Charles Scharf
### Background Sentence
Tim Sloan was succeeded as CEO of Wells Fargo by Charles Scharf.

### Question   [question]   ### Suffix   [answer]   ### Background Sentence

Table 14: **Prompt for Paraphrasing.** A 2-shot prompt for paraphrasing. $y$ indicates the answer for the question and $x$ denotes the remaining part of sentence, as introduced in Section 4.1.

For the following prefix, give me 2 highly diverse paraphrases of the same in high-quality English language as in sentences on Wikipedia. Ensure that the suffix is followed by a paraphrased prefix. Do not inclue numbering. Maintain the sentence structure.
# Sentence
In infrainguinal bypass surgery, the preferred type of graft for optimal outcomes is an Autologous vein.
# Prefix
In infrainguinal bypass surgery, the preferred type of graft for optimal outcomes is an
# Suffix (PRESERVE AND KEEP LETTER CASE)
Autologous vein.
# Paraphrases (Prefix + Suffix)
In infrainguinal bypass procedures, the graft type most recommended for the best results is an Autologous vein.
During infrainguinal bypass operations, the optimal choice for a graft to achieve the best outcomes is an Autologous vein.

For the following prefix, give me 2 highly diverse paraphrases of the same in high-quality English language as in sentences on Wikipedia. Ensure that the suffix is followed by a paraphrased prefix. Do not include numbering. Maintain the sentence structure.
# Sentence
During the embryonic development of the gastrointestinal tract, proper rotation of the gut is necessary for the correct placement of the caecum; an abnormality in this process can lead to Mixed rotation.
# Prefix
During the embryonic development of the gastrointestinal tract, proper rotation of the gut is necessary for the correct placement of the caecum; an abnormality in this process can lead to
# Suffix (PRESERVE AND KEEP LETTER CASE)
Mixed rotation.
# Paraphrases (Prefix + Suffix)
In the formation of the gastrointestinal system during embryonic growth, it is essential for the gut to rotate correctly to ensure the caecum is properly positioned; deviations in this mechanism may result in Mixed rotation.
Throughout the development of the gastrointestinal tract in the embryo, the accurate rotation of the gut is crucial for the appropriate localization of the caecum; any irregularities in this rotation can result in Mixed rotation.

For the following prefix, give me 10 highly diverse paraphrases of the same in high-quality English language as in sentences on Wikipedia. Ensure that the suffix is followed by a paraphrased prefix. Do not include numbering. Maintain the sentence structure.

# Sentence   $(x, y)$   # Prefix   $x$   # Suffix   $y$   # Paraphrases (Prefix + Suffix)

Table 15: Experimental results on datasets with synthetic documents from **diverse LLMs**. We present results from Llama2-7B [48], Mistral-7B-Instruct-v0.2 [18], and Phi3-mini-4k-instruct [1].

| Method | SQuAD-syn | | | StreamingQA-syn | | | ArchivalQA-syn | | |
| | EM | Recall | F1 | EM | Recall | F1 | EM | Recall | F1 |
|---|---|---|---|---|---|---|---|---|---|
| Llama2-7B [48] | | | | | | | | | |
| **No Adaptation** | 17.30 | 25.09 | 24.27 | 29.71 | 36.37 | 35.59 | 15.10 | 23.61 | 22.36 |
| **Fine-Tuning** | 69.10 | 80.34 | 78.09 | **85.30** | 90.97 | 90.57 | 63.60 | 82.54 | 79.26 |
| **FreeLB** [57] | **75.10** | 85.63 | 83.47 | 83.46 | 91.73 | 90.95 | **67.00** | 83.82 | 80.86 |
| **NEFTune** [16] | 71.10 | 84.17 | 81.38 | 82.54 | 90.65 | 89.65 | 64.80 | 82.08 | 79.01 |
| **Ours** | 73.10 | **87.00** | **84.13** | 83.46 | **92.46** | **91.20** | 65.00 | **88.70** | **83.55** |
| Mistral-7B-Instruct-v0.2 [18] | | | | | | | | | |
| **No Adaptation** | 4.90 | 25.33 | 10.86 | 14.70 | 31.78 | 20.58 | 6.60 | 26.66 | 13.37 |
| **Fine-Tuning** | 49.40 | 83.60 | 64.66 | 65.08 | 88.43 | 75.51 | 41.10 | 75.88 | 59.13 |
| **FreeLB** [57] | 58.10 | 86.20 | 71.30 | 72.28 | 93.44 | 82.21 | 47.30 | 82.10 | 66.22 |
| **NEFTune** [16] | 45.10 | 80.06 | 59.67 | 67.84 | 89.01 | 77.34 | 37.70 | 73.25 | 55.58 |
| **Ours** | **73.20** | **89.53** | **83.57** | **83.46** | **94.14** | **91.79** | **64.80** | **89.07** | **82.40** |
| Phi3-mini-4k-instruct [1] | | | | | | | | | |
| **No Adaptation** | 5.20 | 22.20 | 10.77 | 9.95 | 26.88 | 15.21 | 5.20 | 24.84 | 11.06 |
| **Fine-Tuning** | 38.80 | 61.78 | 50.32 | 44.10 | 70.93 | 55.19 | 24.80 | 54.57 | 37.05 |
| **FreeLB** [57] | 41.90 | 62.43 | 52.50 | 50.69 | 72.95 | 60.17 | 22.50 | 57.57 | 35.55 |
| **NEFTune** [16] | 39.30 | 63.70 | 50.52 | 44.87 | 71.54 | 55.87 | 23.60 | 55.95 | 36.40 |
| **Ours** | **53.30** | **67.02** | **62.88** | **70.60** | **77.78** | **75.39** | **30.20** | **64.93** | **47.04** |

# D  Additional Experiments

## D.1  Experiments with Other Language Models

Verifying whether the proposed method can be transferred to other Language Models (LMs) is important. First, we validate our LaPael with Llama-2-7B [48], a non-instruction-tuned version of the Vicuna-7B we used in experiments. In Table 15, we present the experimental results with Llama-2-7B. The results show that our LaPael is effective even in the LM that is not instruction-tuned. In Table 15, we also present the experimental results with Mistral-7B-Instruct-v0.2, which is an instruction-tuned model based on a different LLM Mistral-7B [18]. The results indicate that our LaPael is applicable not only to Llama-based models but also to LMs with different bases. Furthermore, in Table 15, we present the experimental results with Phi3-mini-4k-instruction, which is a pre-trained LLM with 3.8 billion parameters [1]. The results indicate that our LaPael is highly effective when applied to the Phi3-mini model, which has fewer parameters than other LLMs.

## D.2  Experiments with Parameter-Efficient Fine-Tuning

Parameter-efficient fine-tuning is a method that fine-tunes LLMs with minimal updates to their parameters. It is of interest that our LaPael can be effective even with parameter-efficient fine-tuning. LoRA [13] is a well-known method for parameter-efficient fine-tuning, which updates trainable rank decomposition matrices injected into the parameters of LLMs. In Table 16, we present the experimental results with LoRA on Vicuna-7b-v1.5 where we update only the low-rank matrices of `up_proj`, `gate_proj`, and `down_proj` layers. The results demonstrate that LaPael is also effective in LoRA fine-tuning, highlighting its flexible applicability in diverse fine-tuning scenarios.

## D.3  Visualization of Latent Features

In Figure 7, we display the latent features from the final layers of large language models (LLMs) with and without latent paraphrases, where we reduce the dimension using t-SNE [49]. Crosses ('x') mark the embeddings from LLMs with latent paraphrasers. As illustrated in Figure 7, latent paraphrasers enable the generation of diverse data samples, enhancing the diversity compared to data-level paraphrases.

Table 16: Experimental results on datasets with synthetic documents, where we use LoRA [13] instead of fine-tuning full parameters on Vicuna-7b-v1.5 [56].

| Method | SQuAD-syn | | | StreamingQA-syn | | | ArchivalQA-syn | | |
|---|---|---|---|---|---|---|---|---|---|
| | EM | Recall | F1 | EM | Recall | F1 | EM | Recall | F1 |
| **No Adaptation** | 13.10 | 22.91 | 21.09 | 16.39 | 26.30 | 23.71 | 13.50 | 25.07 | 22.12 |
| **Fine-Tuning** | 62.70 | 72.80 | 70.74 | 73.97 | 83.75 | 82.12 | 53.60 | 68.23 | 66.00 |
| **FreeLB** [57] | 62.00 | 77.21 | 73.67 | **81.47** | 88.76 | 87.51 | **62.80** | 77.77 | 74.55 |
| **NEFTune** [16] | **67.40** | 79.10 | 76.59 | 78.71 | 85.77 | 84.65 | 57.60 | 74.88 | 71.35 |
| **Ours** | 65.80 | **82.10** | **78.80** | 80.09 | **89.43** | **88.03** | 61.70 | **79.22** | **75.24** |

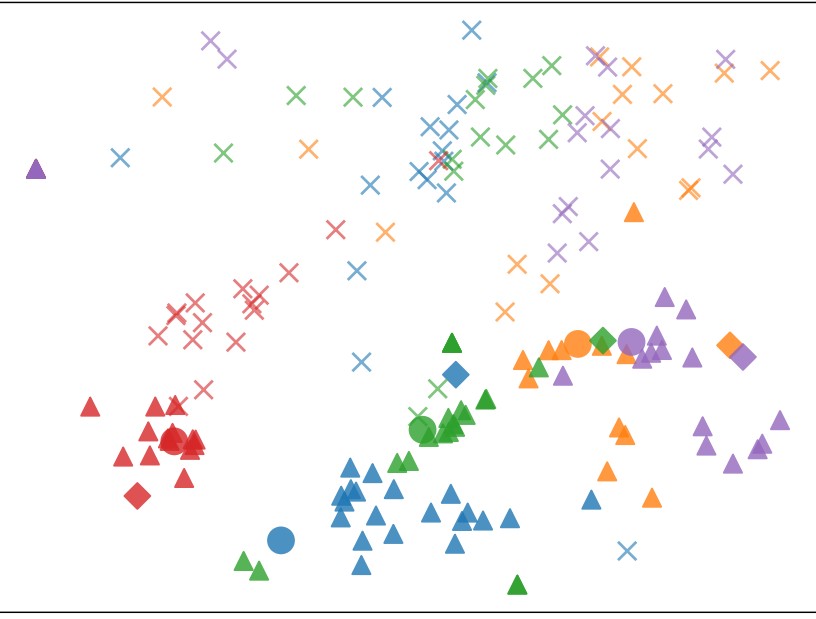

Figure 7: **Visualization of Latent Features.** We visualize the latent features from the last layers of LLMs using 5 randomly sampled data from ArchivalQA dataset. Each color denotes the different data, circles denote the original sentences, triangles denote the paraphrases, diamonds denote the questions, and crosses ('x') denote the original sentence with latent paraphrasing.

