# OpenReview forum: "Latent Paraphrasing: Perturbation on Layers Improves Knowledge Injection in Language Models"
_NeurIPS.cc/2024/Conference — NeurIPS 2024 poster_

### Official Review · Reviewer_NFcZ · 2024-07-02

**Soundness:** 3
**Presentation:** 3
**Contribution:** 2
**Rating:** 5
**Confidence:** 4

**Summary:**

The paper proposes a new method named LaPael to enhance knowledge injection for large language models. Different from traditional data-level augmentations or noise-based methods, LaPael operates at the latent level, preserving the semantic integrity of the text while introducing meaningful variability. Experimental results on three question-answering datasets show that the proposed method outperforms the baselines.

**Strengths:**

1. The authors propose a latent perturbation for enhancing knowledge injection of LLMs, and the proposed method outperforms the baselines.

2. The paper is written-well, with a direct motivation and a well organized structure.

**Weaknesses:**

1. The difference between the method proposed in this paper and the previous perturbation or enhancement methods in the feature space is not explained clearly.

2. Why is the output distribution of Transformer assumed to be Gaussian in Training Part of Section 4.2? What is the effect of directly calculating the KL divergence of two distributions without considering the type of distribution? The author's explanation will help readers better understand the method.

3. The authors claim that the paraphrasing method requires a high computational cost, so they should compare the proposed method with the paraphrasing method on computational cost.

**Questions:**

1. In Table 7, why was it not compared with the Fine Tuning (+para) method?

2. Will the proposed knowledge injection method have a negative impact on the knowledge already mastered by LLMs?

**Limitations:**

Please see above Weaknesses and Questions.

---

> ### Author Rebuttal · Authors · 2024-08-07
>
> We sincerely thank you for your constructive and helpful comments. We initially address all your concerns and questions below.
>
> `[4-1] Difference between the previous peturbation methods`
> > W1. The difference between the method proposed in this paper and the previous perturbation or enhancement methods in the feature space is not explained clearly.
>
> Thank you for giving us a chance to clarify the point. To be clear, we have included the following sentence after Line 109 of the main paper: "Compared to previous perturbation methods, our method significantly differs in terms of the training objective for a perturbation function and its application to the knowledge injection of LLMs. Specifically, previous methods use bi-level optimization to train the perturbation function, while we use KL divergence to optimize the perturbation function by directly matching the output distribution of the perturbed model with that of the model using paraphrased data."
>
> ---
> `[4-2] Clarification on Gaussian used in the output distribution`
> > W2. Why is the output distribution of the Transformer assumed to be Gaussian in the Training Part of Section 4.2? What is the effect of directly calculating the KL divergence of two distributions without considering the type of distribution? The author's explanation will help readers better understand the method.
>
> We appreciate your valuable question. While it is possible to estimate with a more complex distribution, we chose to use a Gaussian distribution due to complexity issues. We found that even with this simplified modeling, our approach performs well.
>
> ---
> `[4-3] Computational cost comparison against paraphrasing method`
> > W3. The authors claim that the paraphrasing method requires a high computational cost, so they should compare the proposed method with the paraphrasing method on computational cost.
>
> Thank you for highlighting this important point. Our method requires less computational cost than the paraphrasing method, as our method requires adding a tiny module on the top of the LLM. To compare computational costs, we calculate each method in FLOPs (Floating Point Operations), with fine-tuning the LLM costing $F_{LLM}$.
>
> 1. **Fine-tuning with paraphrased text**: This requires generating paraphrases, costing $F_{LLM} + F_{Paraphraser}$ FLOPs, where $F_{Paraphraser}$ is the cost of a forward pass on the paraphraser model.
> 2. **Fine-tuning with latent paraphraser**: This requires a forward pass on the latent paraphraser, costing $F_{LLM} + F_{LaPael}$ FLOPs, where $F_{LaPael}$ is the cost of a forward pass on the latent paraphrasers.
>
> A single latent paraphraser has $3d^2 + 4d$ parameters. For a 7B LLM, this is about 50M parameters. Using 5 latent paraphrasers totals 250M parameters. Thus, using a latent paraphraser is **equivalent to a 250M parameter paraphraser**, which is impractical for generating high-quality paraphrases. Given that `gpt-3.5-turbo` (or a 7B LLM) is used as the paraphraser in experiments, our latent paraphraser is significantly cheaper in computational cost.
>
> ---
> `[4-4] Additional experiments for Table 7`
> > Q1. In Table 7, why was it not compared with the Fine Tuning (+para) method?
>
> Thank you for pointing it out. We missed adding FreeLB and Fine-Tuning (+para) in Table 7. We conducted the experiments you mentioned and the results are as follows:
>
> * Table 7 (with FreeLB and Fine-Tuning (+para))
>
> |             | EM    | Recall | F1    |
> | ----------- | ----- | ------ | ----- |
> | No Adapt    | 13.10 | 22.91  | 21.90 |
> | Fine-Tuning | 24.10 | 38.49  | 34.78 |
> | NEFTune     | 22.30 | 39.07  | 33.90 |
> | FreeLB      | 22.30 | 39.95  | 34.31 |
> | Ours        | 27.20 | 46.66  | 39.66 |
> | Fine-Tuning (+para)| 31.90 | 48.13 | 43.08 |
>
> Experimental result shows that using data-level paraphrases is effective in the real-world data setting.
> As we discussed in Section A Limitations, our latent paraphraser cannot resolve the reversal curse issue in the real-world data, while data-level paraphraser can. We have included the results and related discussions in Table 7 of the revision.
>
> ---
> `[4-5] A negative impact on the knowledge retention`
> > Q2. Will the proposed knowledge injection method have a negative impact on the knowledge already mastered by LLMs?
>
> We appreciate your valuable and insightful suggestion. To evaluate the potential negative impact on the knowledge already mastered by the LLM, we used the EntityQuestions dataset [1], which asks simple questions about entities. Among these questions, we only used the simple `place-of-birth` questions from the frequent entities (e.g., Where was Leonardo da Vinci born?), with 988 questions in total.
>
> We fine-tuned the Vicuna-7b-v1.5 model on a synthetic SQuAD document set ($D_K$) using each method and measured the QA performance on EntityQuestions as follows.
>
> * QA Performance on EntityQuestions after fine-tuning LLM on $D_K$ of SQuAD
>
> |             | EM    | Recall | F1    |
> | ----------- | ----- | ------ | ----- |
> | No Adapt    | 59.00 | 64.38  | 63.46 |
> | Fine-Tuning | 52.23 | 55.63  | 55.41 |
> | NEFTune     | 48.28 | 50.33 | 50.54 |
> | FreeLB      | 46.56 | 48.37 | 48.57 |
> | Ours        | 39.88 | 41.97  | 42.12 |
> | Fine-Tuning (+para)| 33.50 | 35.18 | 35.28 |
>
> The experimental results show that all fine-tuning approaches have a negative impact on the knowledge already mastered by the LLM. We also observe that the better the knowledge injection performance, the greater the negative impact on knowledge retention. As our work focuses on improving knowledge injection, knowledge retention is beyond its current scope. However, we believe discussing this point as a limitation is important as it can encourage future work in this direction. We have included this point and the experimental results in the limitations section of our revised manuscript.
>
> ---
> **Reference**
>
> [1] Sciavolino et al., Simple Entity-Centric Questions Challenge Dense Retrievers

---

> > ### Comment · Reviewer_NFcZ · 2024-08-13
> >
> > Thanks to the authors for the detailed responses. After reading the other reviews and the rebuttals, I maintain the original rating.

---

### Official Review · Reviewer_WC5j · 2024-07-11

**Soundness:** 3
**Presentation:** 3
**Contribution:** 3
**Rating:** 6
**Confidence:** 3

**Summary:**

This paper focusses on the ability of LLMs to learn new knowledge. Previous work has shown that paraphrasing techniques help learning this new knowledge. However, as the authors argue, explicit paraphrasing has a high computational cost, and the paraphrased data is of limited diversity. To circumvent this, this paper introduces latent paraphrasing, by adding a latent paraphrasing layer in the standard transformer module, in between the LayerNorm and the MLP. This layer is trained with paraphrased data, generated by an LLM. The optimization objective is to minimize the KL-divergence between the output of a “standard” transformer model with the paraphrases as input, and the output of a transformer including the paraphrasing layer with the original sentences as input.

The authors test their approach, called LaPael, on three datasets: SQuAD, StreamingQA, and ArchivalQA, and compare against multiple fine-tuning baselines. They show that LaPael mostly outperforms the baselines, even in cross-domain transfer experiments.

**Strengths:**

* This paper presents a large number of ablation experiments, that show the benefit of each of the components of the proposed approach. These experiments are very insightful, as at first the method might sound a bit complicated. Moreover, the extensive ablation experiments can help steer future research in this direction.
* The proposed approach is effective, and outperforms the baselines with relatively large margins.

**Weaknesses:**

* The objective of the paper is to explicitly focus on fine-tuning approaches, instead of approaches that rely on external memory. In that light, it makes sense to only compare against fine-tuning baselines. However, if one is just interested in what method performs best on the task, comparing against external memory based approaches would have been a good addition to the results section.
* As a related point, I am wondering to what extent the used datasets can really evaluate new knowledge injection. For example, SQuAD is based on Wikipedia, which the models have probably seen during pretraining. Also see my related question below.
* I also have a few other questions about the approach, which I have added in the question section below.

**Questions:**

* The authors mention that they use Vicuna as the base model. It seems unlikely that the model has never seen the new knowledge represented in D_K during pre-training. At the same time, the authors show that the model without adaptation scores much lower than any of the other baselines. This makes me wonder to what extent the authors think that LaPael measures true new knowledge injection in the LLM?

Some clarifying questions:

* Line 177: regarding the gold mask for tokens that correspond to the named entity. How is determined what the named entities are in the sentence? Do the authors deploy some sort of NER tagger?
* Line 193: mentions “rephrasing each question of D_QA”. Should this be “rephrasing each document”?
* Line 217: mentions that “the LLM is fine-tuned on the test set”. To confirm — does this refer to the paraphrased version of the test set, D_K?
* Line 238: regarding “Combining LaPael and Paraphrases”. I understand the objective of this experiment, and I agree with the authors that it is insightful. I am not entirely sure if I understand the experimental setup here. Specifically, what does “number of paraphrases” mean in the LaPael setting in Figure 4? Is “ours” the default LaPael setup, and “fine-tuning” LaPael + fine-tuning?
* Table 7: Can the authors elaborate on how they adapt the baselines in this real-world data scenario?

**Limitations:**

The limitations are addressed in the appendix.

---

> ### Author Rebuttal · Authors · 2024-08-07
>
> We do appreciate your positive feedback and valuable suggestions, and we hope our response fully addresses your concern.
>
> ---
> `[3-1] Regarding new knowledge injection`
> > W2. As a related point, I am wondering to what extent the used datasets can really evaluate new knowledge injection.
>
> > Q1. It seems unlikely that the model has never seen the new knowledge represented in D_K during pre-training. At the same time, the authors show that the model without adaptation scores much lower than any of the other baselines. This makes me wonder to what extent the authors think that LaPael measures true new knowledge injection in the LLM?
>
> Thank you for your valuable comment. We would like to clarify that we measure the degree of knowledge injection by assessing QA accuracy on the QA dataset ($D_{QA}$), which corresponds to the documents ($D_K$) as defined in Section 3. We consider the knowledge that the LLM fails to answer as new knowledge to the LLM, even if such knowledge was seen during pre-training. Previous works [1, 2] also support these observations, as referenced in Lines 25-29.
>
> However, **we entirely agree with your opinion** on measuring true new knowledge injection. Therefore, we have provided additional experimental results on **two datasets** containing new knowledge to the LLM, with available raw documents (Please see Table A in the attached PDF file on the global comment). The results show that our method is effective with raw documents on new knowledge, confirming the same trend observed in our previous experiments.
>
> We have included these results in Section 5 of the revised manuscript considering their significance.
>
> > Details of the datasets:
> > * Films 2024: A synthetic QA dataset based on raw Wikipedia articles under [2024 films](https://en.wikipedia.org/wiki/Category:2024_films). We generated QAs from wiki documents using GPT-4o following Jiang et al. [3].
> > * Events 2024: A synthetic QA dataset based on raw Wikipedia articles under May, June, and July of [US 2024 events in the United States](https://en.wikipedia.org/wiki/Category:2024_events_in_the_United_States_by_month). We generated QAs from wiki documents using GPT-4o following Ovadia et al. [4].
>
> ---
> `[3-2] Suggestion: Comparison with external memory-based approaches`
> > W1. Comparing against external memory based approaches would have been a good addition to the results section.
>
> As per your great suggestion, we have conducted experiments comparing our approach with Retrieval-Augmented Generation (RAG) on datasets with new knowledge, Films 2024 and Events 2024, as used in the previous answer `3-1`. We use [bge-large-en-v1.5](https://huggingface.co/BAAI/bge-large-en-v1.5) model for the embedding.
>
>
> * Films 2024
>
> |      | EM   | Recall | F1  |
> | ---- | ---- | ---    | ---- |
> | Fine-tuning | 13.39 | 30.03 | 28.84 |
> | Ours | 16.29 | 35.04 | 32.56 |
> | RAG | 48.31  | 72.45 | 67.13 |
>
> * Events 2024
>
> |     | EM   | Recall | F1  |
> | ---- | ---- | ---    | ---- |
> | Fine-tuning | 10.98 | 43.76  | 39.62 |
> | Ours | 15.26 | 56.70 | 46.45 |
> | RAG | 27.17 | 64.02 | 55.71 |
>
> Experimental results show that the memory-based approach (RAG) outperforms the fine-tuning approaches, as observed in Ovadia et al. [4]. However, our method can close the gap between the two approaches, indicating a potential pathway to further improve fine-tuning methods. We have added these experimental results to the revised manuscript to emphasize this point.
>
> ---
> `[3-3] Answers on Questions`
>
> Thank you for your thorough clarification. We have incorporated all clarifications into the revised manuscript. While addressing your questions, we realized that the knowledge injection pipeline with LaPael -- training latent paraphrasers on $D_{train}$ $\rightarrow$ fine-tuning LLMs on $D_{K}$ -- can be confusing. Therefore, we have added a supporting figure to clearly explain that in Section 4 of the revised manuscript.
>
> > Line 177: Question about NER tagger
>
> Yes, we use GPT-3.5-turbo as the NER tagger.
>
> ---
> > Line 193: mentions “rephrasing each question of D_QA”. Should this be “rephrasing each document”?
>
>
> We apologize for the confusion. We would like to clarify that we used **both question and answer** to generate the synthetic document. This is because we want to ensure that the synthetic document contains only relevant information to the question, as illustrated in Table 1.
>
> ---
> > Line 217: mentions that “the LLM is fine-tuned on the test set”. To confirm — does this refer to the paraphrased version of the test set, D_K?
>
> No, it refers to the documents $D_K$ defined in Section 3, not the paraphrased version of $D_K$. We used a paraphrased version of $D_K$ in experiments for Figure 4 and Fine-Tuning (+ para.) in Table 2.
>
> ---
> > Line 238: regarding “Combining LaPael and Paraphrases”. Specifically, what does “number of paraphrases” mean in the LaPael setting in Figure 4? Is “ours” the default LaPael setup, and “fine-tuning” LaPael + fine-tuning?
>
> The *number of paraphrases* in Figure 4 refers to the number of paraphrases for each document in the dataset used to fine-tune the LLM ($D_K$) for both "fine-tuning" and "ours."
>
> ---
> > Table 7: Can the authors elaborate on how they adapt the baselines in this real-world data scenario?
>
> In the real-world data scenario, baselines are also fine-tuned on the raw documents from the SQuAD test set ($D_K$). We would like to clarify that "ours" denotes fine-tuning the LLM on the raw documents from the SQuAD test set ($D_K$), with the latent paraphraser trained on synthetic documents from the SQuAD train set ($D_{train}$).
>
> ---
> **References**
>
> [1] Kandpal et al., Large Language Models Struggle to Learn Long-Tail Knowledge
>
> [2] Allen-Zhu et al., Physics of Language Models: Part 3.1, Knowledge Storage and Extraction
>
> [3] Jiang et al., Instruction-tuned Language Models are Better Knowledge Learners
>
> [4] Ovadia et al., Fine-Tuning or Retrieval? Comparing Knowledge Injection in LLMs

---

> ### Comment · Reviewer_WC5j · 2024-08-09
>
> Thank you for your response, and clarifications. They largely address my concerns, and I appreciate the additional results for the two new datasets.
>
> I have one follow-up question about the results for the new RAG baseline. First of all, thank you for running these experiments, and including the numbers in the rebuttal, and for updating the manuscript with these numbers. Currently, the RAG numbers are quite far from the LaPael numbers. What do you think is the main advantage of using a method like LaPael over the retrieval based method?

---

> > ### Author Response · Authors · 2024-08-10
> >
> > Thank the reviewer for your reply and follow-up question. We are happy to hear that your concerns are largely addressed by our response.
> >
> > Regarding your follow-up question, the main advantage of fine-tuning approaches including ours, over retrieval-based approaches such as RAG lies in simplicity, inference efficiency, and computational cost. Fine-tuning approaches result in a self-contained model that *simplifies* the overall system architecture by eliminating the need for additional infrastructure for document retrieval and ranking during inference. This approach *avoids the extra computational costs* associated with query embedding and extended prompts with retrieved documents, making it more *cost-efficient*, especially in terms of GPU memory, and suitable for deployment in resource-constrained environments.
> >
> > Due to the complexity-performance trade-off, the choice between two approaches depends on the deployment environment. Fine-tuning approaches are specifically suited for settings that require low-latency and resource-efficient deployment. We hope this explanation clarifies your question. If you have any further questions, please let us know.

---

> > > ### Comment · Reviewer_WC5j · 2024-08-12
> > >
> > > Thanks for your elaboration. I appreciate your work for trying a new approach. I'm wondering about the computational cost though, because you also need to train and use the latent paraphraser. I'm assuming the computational cost in the end boils down to certain architectural choices you make. Or would you say LaPael is always computationally more cost efficient than a RAG type of method?

---

> > > > ### Author Response · Authors · 2024-08-13
> > > >
> > > > Thank the reviewer for engaging in this discussion. We are happy to share our response to the question below:
> > > >
> > > > > I'm wondering about the computational cost though, because you also need to train and use the latent paraphraser. I'm assuming the computational cost in the end boils down to certain architectural choices you make. Or would you say LaPael is always computationally more cost efficient than a RAG type of method?
> > > >
> > > > To clarify, **our comparison focused on the computational cost during inference, not training.** In our previous response, we emphasized that fine-tuning methods are more cost-efficient than RAG-based methods in terms of inference cost.
> > > >
> > > > When considering the computational cost for training, fine-tuning methods including LaPael do indeed incur additional costs, particularly due to the fine-tuning of LLMs for knowledge injection. Moreover, LaPael requires further costs for training and utilizing the latent paraphraser, as we mentioned in our responses to Reviewer Zuau (`1-2`) and Reviewer (`4-3`).
> > > >
> > > > We hope our answer clarifies the inquiry. If there are any further questions, please let us know.

---

> > > > > ### Comment · Reviewer_WC5j · 2024-08-13
> > > > >
> > > > > Thank you! I think it would be valuable to add some additional discussion around this topic in the paper, as it clarifies the motivation for your approach.

---

> > > > > > ### Author Response · Authors · 2024-08-14
> > > > > >
> > > > > > We do appreciate your valuable suggestions on our work and will gladly continue improving our paper. Thank the reviewer once again for your time and effort in helping us enhance our work.

---

### Official Review · Reviewer_bjHu · 2024-07-11

**Soundness:** 1
**Presentation:** 1
**Contribution:** 2
**Rating:** 3
**Confidence:** 5

**Summary:**

This paper presents LaPael, a novel approach to injecting new knowledge into Large Language Models (LLMs). Previous works have shown that fine-tuning the model with data augmented by paraphrasing helps the model learn new knowledge. However, this requires high-quality paraphrased data each time new knowledge is injected, which is costly. The paper proposes a method that introduces a paraphraser layer within the LLMs, which acts as a latent paraphraser that paraphrases the input text in the latent space. The authors train the latent paraphraser using paraphrased data. Once the latent paraphraser is trained, it is used to fine-tune the LLM on the new knowledge. LaPael outperforms standard fine-tuning and paraphrasing approaches, showing improvements on question-answering benchmarks.

**Strengths:**

1. The paraphraser layer within LLMs introduces a novel method for incorporating new knowledge, reducing the need for high-quality paraphrased data and reducing the significant cost of fine-tuning the model.

2. The paper demonstrates that LaPael outperforms the supervised fine-tuning method on question-answering benchmarks.

**Weaknesses:**

**The major weakness of the paper lies in the experiment section and the details provided regarding the experiment. Below are my concerns and questions related to this section:**

1. Line 190: "We use the subset of questions in the test set of each dataset for $D_{QA}$" -> Why is only a subset of questions chosen instead of the entire test set? No details about this are mentioned in the paper.

2. Creating a document out of question-and-answer pairs to inject knowledge into the LLM and then testing it on the same test set is inappropriate for evaluating knowledge injection. It would have been more suitable to use datasets with available documents, inject the knowledge from these documents, and then test the model on the QA. Additionally, the documents used should not have been seen by the pretrained LLM during its pretraining phase to quantify how much knowledge has been injected into the LLM.

3. There are no statistics available for the created documents, such as their size. Creating small paragraphs from question-and-answer pairs, fine-tuning the model on these paragraphs, and then testing it on the same question-and-answer pairs does not constitute a correct and fair evaluation.

4. Line 212:  Why are in-context examples used instead of experimenting with a zero-shot setting to assess the extent of the model's learned knowledge?

5. Lines 184-185: It is not clear what the authors meant by "We sample N noise."
6. Line 182: "From Equation 10" should be "from Equation 15."

**The writing of the paper is also very weak. Here are a few examples:**

a. The acronym "LM" is used in the experiment section without being defined. The paper uses "LLM" for Large Language Models and sometimes "LM."

b. The paper uses "$D_{knowledge}$" and "$D_{K}$" interchangeably. Please ensure consistency.

c. There are missing details in the experiment section, and no correct reference is provided to indicate where in the appendix these details can be found.

**Questions:**

Please refer to the weaknesses section.

---

> ### Author Rebuttal · Authors · 2024-08-07
>
> We sincerely appreciate your valuable comments on our work. We understand your concerns and are certain that your comments will help improve the quality of our work. Should you have any unresolved concerns, please let us know. We are happy to discuss and do our best to address your concerns.
>
> ---
>
> `[2-1] Experiments with available documents with new knowledge.`
> > W2. It would have been more suitable to use datasets with available documents. Additionally, the documents used should not have been seen by the pretrained LLM during its pretraining phase to quantify how much knowledge has been injected into the LLM.
>
> **Summary: We show that our method is still effective with raw documents and new knowledge by conducting supplementary experiments on four datasets. The experimental results are presented in Table A of the attachment.**
>
> Thank you for your valuable suggestion. We agree that experiments on datasets with available raw documents and new knowledge are important. We acknowledge that the scope of existing experiments, including Table 7, is limited as they do not handle the new knowledge scenario.
>
> To address this, we conducted new experiments on four datasets, including two with new knowledge, using available raw documents. Specifically, SQuAD-raw and StreamingQA-raw analyze knowledge injection with widely-used document-question datasets, while Films 2024 and Events 2024 test new knowledge injection, as Vicuna-7b-v1.5 is not pre-trained on them (see below for the details of the datasets).
>
> Table A in the attachment on the global comment presents the experimental results, showing our method's effectiveness with available raw documents and new knowledge, consistent with previous synthetic document experiments in the paper. We have included these results in Section 5 of the revised manuscript considering their significance.
>
> > Details of the datasets:
> > * SQuAD-raw: The **entire set** of SQuAD test set with available raw documents.
> > * StreamingQA-raw: The **entire set** of StreamingQA test set with available raw documents.
> > * Films 2024: A synthetic QA dataset based on raw Wikipedia articles under [2024 films](https://en.wikipedia.org/wiki/Category:2024_films). We generated QAs from wiki documents using GPT-4o following Jiang et al. [1].
> > * Events 2024: A synthetic QA dataset based on raw Wikipedia articles under May, June, and July of [US 2024 events in the United States](https://en.wikipedia.org/wiki/Category:2024_events_in_the_United_States_by_month). We generated QAs from wiki documents using GPT-4o following Ovadia et al. [2].
>
> ---
>
> `[2-2] The explanations on why we create a synthetic document out of QA pairs to evaluate the knowledge injection`
> > W1. Why is only a subset of questions chosen instead of the entire test set? No details about this are mentioned in the paper.
>
> > W2. Creating a document out of question-and-answer pairs to inject knowledge into the LLM and then testing it on the same test set is inappropriate for evaluating knowledge injection.
>
> We would like to clarify that we use datasets with synthetic documents to simplify the evaluation and analysis of knowledge injection, based on widely-used QA datasets. We used a subset of them to meet the budget limits on costs for `gpt-4-turbo` API calls and LLM fine-tuning. We constructed synthetic documents from existing question-and-answer pairs to ensure learning from these documents leads to knowledge injection, without considering variables like reversal curse and distracting contents, as mentioned in Line 193. We have clarified these points in the revision.
>
> ---
>
> `[2-3] The statistics for the documents used in experiments.`
> > W3. There are no statistics available for the created documents, such as their size.
>
> Thank you for pointing it out. We missed referencing the statistics initially, although they were in Table 8 of the Appendix. We have corrected this in the revision.
>
> We also supplemented Table 8 with dataset size measurements and presented these in Table B of the attachment in the global comment. Additionally, we measured token counts per document and QA pair for all datasets used, plotting the histogram in Figure A of the attachment. These statistics are now included in Appendix B.1 of the revised manuscript.
>
> ---
>
> `[2-4] Clarification on in-context examples used.`
> > W4. Line 212: Why are in-context examples used instead of experimenting with a zero-shot setting to assess the extent of the model's learned knowledge?
>
> We use in-context examples to ensure that LLMs generate the answer in the desired format (phrase instead of sentence). We have clarified this point in Section 5.1 of the revision.
>
> ---
>
> `[2-5] Regarding the noise sampling in lines 184-185.`
> > W5. Lines 184-185: It is not clear what the authors meant by "We sample N noise."
>
> We would like to clarify that "We sample N noise" means that we randomly sample N different $\alpha$ values from the Gaussian distribution with mean $\mu$ as defined in Equation 5. We have clarified this point in the revision.
>
> ---
> `[2-6] Writing fixations.`
> > W6. Line 182: "From Equation 10" should be "from Equation 15."
>
> > Wa. The acronym "LM" is used in the experiment section without being defined. The paper uses "LLM" for Large Language Models and sometimes "LM."
>
> > Wb. The paper uses "$D_{knowledge}$" and "$D_K$" interchangeably. Please ensure consistency.
>
> > Wc. There are missing details in the experiment section, and no correct reference is provided to indicate where in the appendix these details can be found.
>
> Thank you for pointing those out. We have ensured consistent use of the acronym "LLM" and the term "D_K" throughout the paper in the revision. We have also fixed the no-reference issue by adding the references to the correct subsection of the Appendix.
>
> ---
> **References**
>
> [1] Jiang et al., Instruction-tuned Language Models are Better Knowledge Learners
>
> [2] Ovadia et al., Fine-Tuning or Retrieval? Comparing Knowledge Injection in LLMs

---

> > ### Comment · Reviewer_bjHu · 2024-08-12
> >
> > Thank you to the authors for their rebuttal. I have read the authors' responses and the comments from other reviewers.

---

> > > ### Author Response · Authors · 2024-08-13
> > >
> > > We would like to thank the reviewer for their engagement during this phase. We hope we have addressed the issues raised. It would be very helpful if the reviewer could provide any further concerns or suggestions that remain.
> > > Thank the reviewer once again for your time and effort in this process.

---

### Official Review · Reviewer_Zuau · 2024-07-13

**Soundness:** 3
**Presentation:** 3
**Contribution:** 2
**Rating:** 5
**Confidence:** 4

**Summary:**

This paper proposed a latent paraphraser to generate paraphrased data which will be used as augmented data in LLMs' fine-tuning.
To tackle the challenges of repetitive external model interventions, the latent paraphraser (LaPael) is trained to add a small perturbation at the latent feature level of the LLM and eliminate the need for users to repeatedly paraphrase using LLMs once latent paraphrasers are trained. Specially, with the paired-data as input, i,e., original and paraphrased sentence, the training objective is to minimise the KL divergence between distributions of original sentence after the paraphraser transformation and the distribution of the paraphrased sentence. The experimental results show that LaPael outperforms existing noise-based paraphraser in generating better augmentation data.

**Strengths:**

1. The paper is well writen and easy-to-follow and the research topic of generating augmentation data is important.
2. The empirical results clearly show the superiority of the proposed method, LaPael.

**Weaknesses:**

1. The cost of training the latent paraphraser is not clear, as expensive training will impedes the efficiency of the proposed method.
2. Lack of qualitative analysis of show how the phrased sentences differ from the given paraphrased sentences from GPT3.5.

**Questions:**

1. In table3, the cross-domain transfer: why the results trained on X evaluated on X, is worse than trained on X but evaluated on Y.
2. Any explainations or case study showing the generated paraphase sentences from LaPael?
3. The computation cost comparsion with noise-based paraphraser.

**Limitations:**

The trade-off between task performance and computation cost.

---

> ### Author Rebuttal · Authors · 2024-08-07
>
> We sincerely thank you for your constructive and helpful comments. We initially address all your concerns and questions below.
>
> ---
> `[1-1] Regarding the comparative analysis of paraphrased sentences`
> > W2. Lack of qualitative analysis of show how the phrased sentences differ from the given paraphrased sentences from GPT3.5.
>
> > Q2. Any explanations or case studies showing the generated paraphrase sentences from LaPael?
>
> Thank you for your feedback. We would like to clarify our approach as follows. Instead of generating paraphrased sentences, our proposed method perturbs the latent representations within the LLM, as denoted in Lines 45-46. These perturbed latent representations do not directly correspond to paraphrased sentences. Therefore, comparing our method to paraphrased sentences from GPT-3.5 is not applicable. We hope this clarification addresses your concern.
>
> ---
> `[1-2] Regarding the cost of training the latent paraphraser`
> > W1. The cost of training the latent paraphraser is not clear, as expensive training will impede the efficiency of the proposed method.
>
> > Q3. The computation cost comparsion with noise-based paraphraser
>
> Thank you for your valuable feedback and questions. Our proposed method does introduce an additional cost for training the latent paraphraser once prior to fine-tuning an LLM, compared to noise-based fine-tuning baselines. However, we would like to emphasize that this cost becomes negligible over multiple knowledge injections due to the following factors:
>
> 1. **Size and Parameter Cost**: The latent paraphraser has only 3.6% of the parameters of the main LLM (we used a 7B model), keeping the training cost low.
>    - **FLOPs Comparison**: A single forward step for the LLM costs $F$ FLOPs (Floating Point Operations). Training the five latent paraphrasers requires approximately $1.108F$ FLOPs, compared to the $3F$ FLOPs needed for updating the LLM.
>
> 2. **Training Dataset Size**: The latent paraphraser can be effectively trained with a small dataset. Training with 50-100 documents is sufficient to surpass baseline performance, reducing the number of training steps and data required, as shown in Figure 5(a).
>
> 3. **Transferability**: The latent paraphraser is trained only once and can be reused over multiple knowledge injections without additional retraining, as shown in Table 3. As more knowledge is injected, the amortized training cost approaches zero.
>
> We have clearly stated the cost of training the latent paraphraser compared to noise-based fine-tuning baselines in the revised manuscript.
>
> ---
>
> `[1-3] Regarding the cross-domain transfer`
> > Q1. In table3, the cross-domain transfer: why the results trained on X evaluated on X, is worse than trained on X but evaluated on Y.
>
> Thank you for pointing this out. We would like to clarify the document on which the latent paraphraser is trained ($D_{train}$) differs from the document on which the LLM is fine-tuned ($D_{K}$). Therefore, it is possible that the latent paraphraser trained on $D_{train}$ of X can show better performance on $D_K$ of Y than on $D_K$ of X. Moreover, the performance differences are marginal, indicating almost no significant difference. This highlights the transferability of our method, demonstrating that the training dataset for the latent paraphraser does not significantly affect its performance. This flexibility is shown in Table 3 and described in Lines 230-238.

---

> > ### Comment · Reviewer_Zuau · 2024-08-10
> >
> > Thanks very much for your detailed clarification!
> >
> > Follow up questions:
> > (1) Can you give a comparison here between noise-based paraphraser (such as FreeLB) and LaPael?
> > (2) I am curious, after adding such a latent Paraphraser layer, what are the inference performances changes on Unrelated but Basic Reasoning datasets?  such as math reasoning, code generation. As the inplementation is expected to not degrader the basic model capacity.

---

> > > ### Author Response · Authors · 2024-08-11
> > >
> > > Thank you for reviewing our response and providing additional comments. We greatly appreciate your feedback and will address your questions below.
> > >
> > > ---
> > > > Can you give a comparison here between noise-based paraphraser (such as FreeLB) and Lapael?
> > >
> > > Here is the comparison between two noise-based baselines and Lapael. The noise-based baselines, NEFTune and FreeLB, do not require a preliminary step prior to fine-tuning LLMs:
> > >
> > > * **NEFTune:** During fine-tuning, NEFTune adds random noise to token embeddings during fine-tuning to improve model robustness. [1]
> > > * **FreeLB:** During fine-tuning, FreeLB introduces adversarial perturbations to token embeddings, aiming to minimize adversarial risk by optimizing the model's performances across various perturbations. [2]
> > > * **LaPael (ours):** Uses an *input-dependent noise generator* (latent paraphrasers) that applies noise to latent features, learning the noise distribution directly from given paraphrases. LaPael optimizes its latent pararphrasers by aligning the output distributions between the perturbed model with the original sentence and the model with paraphrased sentences, as detailed in Section 4.2 of the main paper. This approach enables LaPael to generate noise that is more contextually appropriate and effective.
> > >
> > > We would like to emphasize that while LaPael requires an additional step of training latent paraphrasers before fine-tuning the LLM, this overhead becomes negligible (previous response `1-2`) and significant performance gains can be achieved compared to the baselines (Tables 2 and 3). We have added this detailed comparison in addition to the content on Lines 105-107 of the main manuscript.
> > >
> > > ---
> > > > I am curious, after adding such a latent Paraphraser layer, what are the inference performance changes on Unrelated but Basic Reasoning datasets? such as math reasoning, code generation. As the implementation is expected to not degrader the basic model capacity.
> > >
> > > Thank you for the insightful question. Fine-tuning LLMs for knowledge injection can indeed degrade the model's capacity due to *catastrophic forgetting* induced by the fine-tuning process. To illustrate this, we provide the math reasoning performance of the LLM (fine-tuned on the synthetic documents of the SQuAD dataset) on the GSM8K [3] test set.
> > >
> > > * Performance on GSM8K test set after knowledge injection
> > >
> > > | GSM8K    | Accuracy |
> > > | -------- | -------- |
> > > | No adapt.| 23.05    |
> > > | Fine-Tuning| 18.88  |
> > > | Ours (Fine-Tuning with LaPael) | 18.04  |
> > >
> > > The results show that fine-tuning itself largely contributes to the degradation of math reasoning performance. As discussed in our response to Reviewer NFcZ (`4-5`), maintaining reasoning performance after fine-tuning is beyond the scope of our work, which primarily focuses on improving knowledge injection. Nonetheless, we acknowledge the importance of addressing this problem and its potential to guide future research. We have included this point and the associated results in the limitations section of our revised manuscript.
> > >
> > > ---
> > > We hope that our response sufficiently addresses your questions. Should you have any further inquiries or comments, please don’t hesitate to let us know so that we can resolve them. If you find that most of your concerns have been addressed, we kindly ask that you consider adjusting the score accordingly. Thank you so much for your effort and time in reviewing our work.
> > >
> > > **Reference**
> > >
> > > [1] Jain et al., NEFTune: Noisy Embeddings Improve Instruction Fine-tuning.
> > >
> > > [2] Zhu et al., FreeLB: Enhanced Adversarial Training for Natural Language Understanding.
> > >
> > > [3] Cobbe et al., Training Verifiers to Solve Math Word Problems

---

> ### Comment · Reviewer_Zuau · 2024-08-12
>
> Q1: I am asking for the **computation cost comparison** between noise-based and the proposed method, as the author mentioned "We have clearly stated the cost of training the latent paraphraser compared to noise-based fine-tuning baselines in the revised manuscript.", But I couldn't find it.
>
> Q2: Thanks for your results and the acknowledgment of "**this degradation of math reasoning performance**".
>
> I keep my original score mostly because the inefficiency in keeping the (other) basic reasoning capability, e.g., math of the LLMs after inserting such paraphrase layer, as one of the important principle of knowledge injecting is not to destroy the existing knowledge capacity.

---

> > ### Author Response · Authors · 2024-08-13
> >
> > Thank the reviewer for clarifying the questions.
> >
> > > Q1: I am asking for the computation cost comparison between noise-based and the proposed method, as the author mentioned "We have clearly stated the cost of training the latent paraphraser compared to noise-based fine-tuning baselines in the revised manuscript.", But I couldn't find it.
> >
> > We apologize for any confusion. Regarding the revised manuscript, we have consistently addressed all comments in the revised version of the manuscript, which unfortunately we cannot upload at this moment.
> >
> > Here, we provide the computational cost comparison between noise-based baselines and the proposed method in terms of per-step FLOPs for training:
> >
> > ### Noise-based baselines:
> > * Fine-tuning LLM: $3F$ FLOPs
> > ### The proposed method:
> > * Fine-tuning LLM: $3.036F$ FLOPs
> > * Additional training for latent pararphasers: $1.108F$ FLOPs.
> >
> > We believe that our method offers a viable option to **enhance the effectiveness of knowledge injection** into LLMs, despite requiring some initial training cost. This cost leads to superior results compared to noise-based baselines.
> >
> > > Q2: Thanks for your results and the acknowledgment of "this degradation of math reasoning performance". I keep my original score mostly because the inefficiency in keeping the (other) basic reasoning capability, e.g., math of the LLMs after inserting such paraphrase layer, as one of the important principle of knowledge injecting is not to destroy the existing knowledge capacity.
> >
> > Thank the reviewer for sharing valuable thoughts.
> > While we understand the concern regarding fine-tuning and catastrophic forgetting, we would like to emphasize that **this is a common challenge across all fine-tuning methods, not just our approach**, as evidenced by the previous work [1].
> > Our method does not significantly exacerbate this issue while improving the knowledge injection performance. From this perspective, we believe our method offers a valuable contribution.
> >
> > We are very grateful for the opportunity to discuss these points with the reviewer. The reviewer's engagement in this discussion is highly valuable to us. Please let us know before the discussion period ends if there are any further concerns or questions.
> >
> > **Reference**
> >
> > [1] Jang et al., Towards Continual Knowledge Learning of Language Models

---

### Author Rebuttal · Authors · 2024-08-07

(R1=R-Zuau, R2=R-bjHu, R3=R-WC5j, R4=R-NFcZ)

We sincerely thank the reviewers for their thoughtful and constructive feedback. We appreciate the acknowledgment that the paper is well-written and organized (R1, R4), the superiority of the proposed method (R1, R2, R3, R4), the insightfulness of the experiments (R3), and the novelty of the method (R2).

Regarding the concerns and questions raised, we believe that we have adequately addressed each one and provided detailed responses in line with each review. We have also revised the manuscript according to your valuable feedback and suggestions.

In the attachment, we include the following:
- Table A: Experimental results on four datasets with raw documents, referenced in the responses to R2 and R3.
- Figure A: The distributions of token counts in documents, questions, and answers for each dataset used in our experiments, referenced in the response to R2.
- Table B: The size of datasets used in our experiments, referenced in the response to R2.

---

### Decision · Program_Chairs · 2024-09-25

**Decision:**

Accept (poster)

**Comment:**

The paper noticed that during knowledge injection in LLMs, data augmentation by paraphrasing is very important. And inspired by this, they proposed Latent Paraphrasing (LaPael) as a novel approach by introducing perturbations at the latent feature level, rather than relying solely on data-level paraphrasing. When injecting new knowledge into an LLM, LaPael first insert a learnable noisy mask before the MLP layers in Transformers and train the mask distribution on an existing paraphrase dataset. Then during the fine-tuning process, this mask distribution induces variability similar to paraphrased data but operates directly within the model's latent space. The paper presents strong experimental results for LaPael, supported by improvements over data augmentation in performance across various benchmarks, including question-answering tasks.

The reviewers have pointed out some concerns that should be addressed. A major concern is the lack of detailed analysis regarding the computational cost of training the latent paraphraser, which could potentially limit the practical applicability of the method. Furthermore, the paper would benefit from additional qualitative analysis to illustrate how the perturbations at the latent level differ from traditional paraphrasing at the data level. There are also questions regarding the generalizability of LaPael across different task, but not just question-answering.

However, the reviewers generally agree that LaPael offers a novel approach by targeting the latent feature space, which is a promising direction that could reduce dependency on external paraphrasing models and improve the knowledge injection process. The paper also provides a potentially new viewpoint of the training dynamics of LLMs, where at the end of training, the model needs some proper weight perturbation to learn new knowledge. Overall the Pros trumps the Cons, which lead to a recommendation to acceptance, but I still recommend authors to put significant efforts to resolve the concerns.